# POPULATION TRANSFORMER: LEARNING POPULATION-LEVEL REPRESENTATIONS OF NEURAL ACTIVITY

**Geeling Chau**[1*]  **Christopher Wang**[2*]  **Sabera Talukder**[1]  **Vighnesh Subramaniam**[2]

**Saraswati Soedarmadji**[1]  **Yisong Yue**[1]  **Boris Katz**[2]  **Andrei Barbu**[2]

[1]California Institute of Technology
{gchau, sabera, ssoedarm, yyue}@caltech.edu
[2]MIT CSAIL, CBMM
{czw, vsub851, boris, abarbu}@mit.edu

## ABSTRACT

We present a self-supervised framework that learns population-level codes for arbitrary ensembles of neural recordings at scale. We address key challenges in scaling models with neural time-series data, namely, sparse and variable electrode distribution across subjects and datasets. The Population Transformer (PopT) stacks on top of pretrained temporal embeddings and enhances downstream decoding by enabling learned aggregation of multiple spatially-sparse data channels. The pretrained PopT lowers the amount of data required for downstream decoding experiments, while increasing accuracy, even on held-out subjects and tasks. Compared to end-to-end methods, this approach is computationally lightweight, while achieving similar or better decoding performance. We further show how our framework is generalizable to multiple time-series embeddings and neural data modalities. Beyond decoding, we interpret the pretrained and fine-tuned PopT models to show how they can be used to extract neuroscience insights from large amounts of data. We release our code as well as a pretrained PopT to enable off-the-shelf improvements in multi-channel intracranial data decoding and interpretability. Code is available at https://github.com/czlwang/PopulationTransformer.

## 1 INTRODUCTION

Building effective representations of neural data is an important tool in enabling neuroscience research. Recordings from the brain such as intracranial (iEEG) and scalp (EEG) electroencepholography, consist of time series recorded simultaneously from multiple channels. The relationships between these time series are complex and governed by the underlying functional connectivity that exists between brain regions. Our goal is to build an effective model of multi-channel activity. Recently, improvements have been made in modeling time-series (Wang et al., 2022; Talukder et al., 2024; Yue et al., 2022; Ansari et al., 2024). This suggests an approach for learning multi-channel representations via aggregating temporal embeddings. However, this is not a trivial task. For brain recordings, particularly iEEG, one must contend with sparse and variable electrode layouts, which change the semantics of input channels from subject to subject. This forces many Brain Machine Interface (BMI) and neuroscience studies to rely on expensive schemes, in which models are retrained for each new participant, requiring large amounts of data for calibration (Faezi et al., 2021; Herff et al., 2020; Martin et al., 2018; Metzger et al., 2023; Willett et al., 2023). To this end, we propose a self-supervised learning framework, Population Transformer (PopT), which is specifically designed to aggregate single-channel encodings across variable electrode layouts.

Self-supervised pretraining on unannotated data has been shown to be effective for creating generic representations that are useful for many downstream tasks (Bommasani et al., 2022). Prior work has shown how to pretrain subject-specific (Le & Shlizerman, 2022) or channel-specific (Wang et al., 2022) models of iEEG, but such techniques ignore inter-channel relationships or commonalities that

---

[*]Equal contribution. Code: https://github.com/czlwang/PopulationTransformer

might exist across subjects. Recent end-to-end self-supervised learning approaches downsample signals heavily to make training across hundreds of channels feasible (Zhang et al., 2024; Yang et al., 2024; Jiang et al., 2024). This is particularly problematic for high-fidelity iEEG signals, which capture sub-millisecond changes in neural activity. Our approach leverages existing rich temporal embeddings to represent signal, freeing the model to focus on learning effective aggregation.

We propose Population Transformer (PopT), a self-supervised pretraining approach that learns subject-generic representations of arbitrary electrode ensembles. Transformers offer the flexibility to learn aggregate information across channels, but large amounts of data are needed to train the attention weights (Devlin et al., 2019). During pretraining, we train on large amounts of unannotated data and simultaneously optimize both a channel-level and ensemble-level objective. These tasks encourage the model to develop subject-generic representations by modeling (1) individual channels in the context of surrounding channels and (2) channel ensembles and their relations across time.

Our PopT approach is modular, and builds on top of powerful single-channel temporal embeddings, which provides two key advantages. First, by separating single-channel embedding and multi-channel-aggregation into different modules, we make our approach agnostic to the specific type of temporal embedding used, leaving room for future independent improvements along either the temporal or spatial dimension (an approach that has been validated in video modeling (Arnab et al., 2021)). Second, by taking advantage of learned channel embeddings, PopT training is computationally lightweight compared to their end-to-end counterparts (Appendix B) and baseline aggregation approaches (Figure 4), allowing for adoption in lower compute resource environments.

Empirically, we find that our pretrained PopT outperforms commonly used aggregation approaches (Ghosal & Abbasi-Asl, 2021), and is competitive with end-to-end trained methods (Zhang et al., 2024; Yang et al., 2024; You et al., 2019). Moreover, we find that these benefits hold even for subjects not seen during pretraining, indicating its usefulness for new subject decoding. We also show that the pretrained PopT weights themselves reveal interpretable patterns for neuroscientific study. Finally, we demonstrate that our proposed framework is agnostic to the underlying temporal encoder, further allowing it to adapt to other neural recording modalities.

Our main contributions are:

1. a generic self-supervised learning framework, Population Transformer (PopT), that learns joint representations of arbitrary channel ensembles across neural datasets,
2. a demonstration that pretraining systematically improves ensemble representations for downstream decoding even for held-out subjects,
3. a new method for brain region connectivity analysis and functional brain region identification based on the pretrained and fine-tuned PopT weights,
4. a trained and usable off-the-shelf model that computes population-level representations of high temporal resolution intracranial neural recordings.

## 2 RELATED WORK

**Self-supervised learning on neural data**  Channel independent pretrained models are a popular approach for neural spiking data (Liu et al., 2022), intracranial brain data (Wang et al., 2022; Talukder & Gkioxari, 2023), and general time-series (Talukder et al., 2024). Additionally, in fixed-channel neural datasets, approaches exist for EEG (Chien et al., 2022; Kostas et al., 2021; Yi et al., 2023), fMRI (Thomas et al., 2022; Kan et al., 2022; Ortega Caro et al., 2023), and calcium imaging (Antoniades et al., 2023) datasets. However, these approaches do not learn population-level interactions across datasets with different recording layouts, either due to a single-channel focus or the assumption that the channel layout is fixed. Several works pretrain spatial and temporal dimensions across datasets with variable inputs (Zhang et al., 2024; Yang et al., 2024; Jiang et al., 2024; Ye et al., 2024; Cai et al., 2023), but most simultaneously learn the temporal embeddings with the spatial modeling, which make them challenging to interpret and computationally expensive to train, especially for high temporal resolution signals. To our knowledge, we are the first to study the problem of building pretrained channel aggregation models on top of pre-existing temporal embeddings trained across neural datasets with variable channel layouts, allowing for modeling of high quality neural data.

**Modeling across variable input channels**  Modeling spatial representations on top of temporal embeddings has been found to be beneficial for decoding (Faezi et al., 2021; Le & Shlizerman,

2022; Azabou et al., 2024), but prior works use supervised labels, so do not leverage large amounts of unannotated data. The brain-computer-interface field has studied how to align latent spaces (Pandarinath et al., 2018; Karpowicz et al., 2022; Degenhart et al., 2020; Jude et al.; Ma et al., 2023) which either still requires creating an alignment matrix to learn across datasets or only provides post-training alignment mechanisms rather than learning across datasets. Other approaches impute missing channels or learn latent spaces robust to missing channels (Talukder et al., 2022; Zhang et al., 2021; Chau et al., 2024), but these are more suited for the occasional missing channel rather than largely varying sensor layouts. We directly learn spatial-level representations using self-supervised learning across datasets to leverage large amounts of unannotated intracranial data.

## 3 POPULATION TRANSFORMER APPROACH

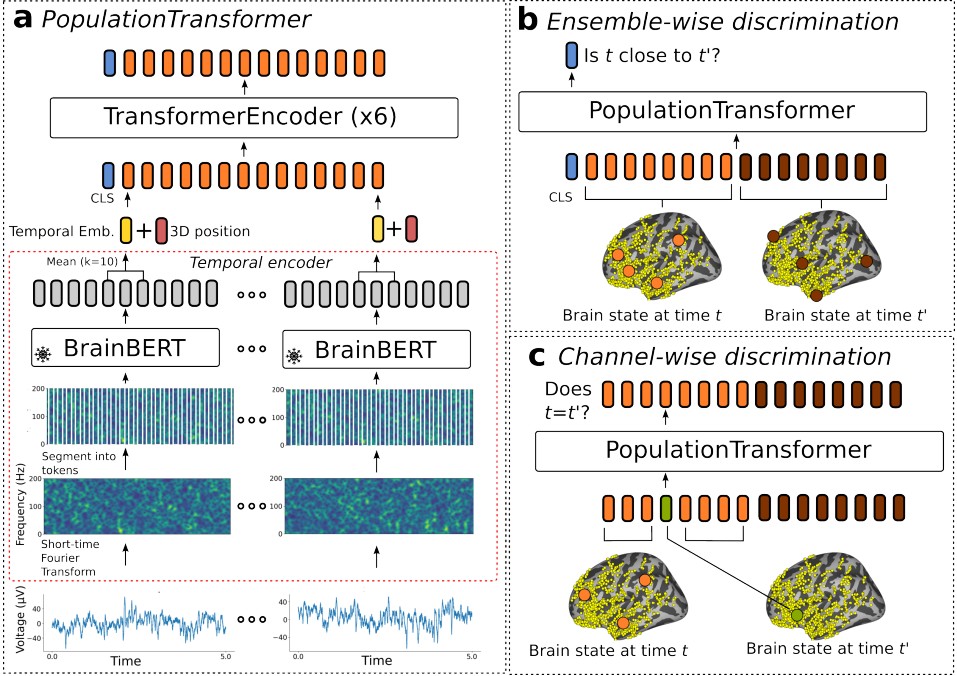

Figure 1: **Schematic of our approach**. The inputs to our model (a) are the neural activities from a collection of electrodes in a given time interval (bottom). These are passed to a frozen temporal embedding model (dotted red outline: BrainBERT (Wang et al., 2022) shown), which produces a set of time embedding vectors (yellow). The 3D positions of each electrode (red) are summed with these vectors to produce the model inputs (orange, lower). PopT produces space-contextual embeddings (orange, top) for each electrode and a `[CLS]` token (blue, top), which can be fine-tuned for downstream tasks. In pretraining, PopT learns two objectives simultaneously. In the first, (b) PopT determines whether two different sets of electrodes (orange vs brown) represent consecutive or non-consecutive times. In the second objective, (c) PopT must determine whether an input channel has been replaced with activity at a random other time that is inconsistent with the majority of inputs.

Figure 1 overviews our Population Transformer (PopT) approach. The key ideas are: (1) to learn a generic representation of neural recordings that can handle arbitrary electrode ensembles; and (2) to employ a modular system design that uses a transformer architecture to aggregate information from existing per-channel temporal embeddings. To do so, we employ a self-supervised pretraining approach to learn ensemble and channel level representations. Afterwards, one can fine-tune PopT on downstream decoding tasks. In addition to offering strong decoding results, including generalization to new subjects with different electrode configurations than training subjects (see Section 5), the modular system design is computationally lightweight (see Appendix B), can benefit from improved temporal representations, and is more readily interpretable (see Section 6).

**Architecture** A schematic of our Population Transformer (PopT) approach is shown in Figure 1. We adopt a transformer backbone due to its ability to accommodate variable channel configurations.

Consider a given subject with $N$ channels indexed by $C = \{1, ..., N_c\}$, and an arbitrary subset of channels $S \subseteq C$. Let $x_i^t \in \mathbb{R}^T$ denote a time window of activity from channel $i$ that begins at time $t$, where $T$ is the number of time samples in the interval. The PopT takes as input a collection of such channels activities, $X^t = \{x_i^t | i \in S\}$, as well as a special [CLS] token. Per channel, each interval of brain activity is passed through a temporal embedding model $B$, in the figure's case BrainBERT (Wang et al., 2022), to obtain a representation of each channel's temporal context, $B(x_i^t) \in \mathbb{R}^d$, where $d$ is the embedding dimension. For BrainBERT, the first step of pre-processing involves obtaining the STFT for the signal, but preprocessing will differ depending on the embedding model used.

To allow the model to learn a common brain state representation across layouts, each channel's embedding is summed with its 3D position, so that the final processed input to the PopT is $X_B^t = \{B(x_i^t) + pos(i) | x_i^t \in X^t\}$. The PopT receives this as an $S \times d$ matrix. Spatial location is given by the electrode's Left, Posterior, and Inferior coordinates for iEEG electrodes (Wideman, 2024), and XYZ positions for EEG electrodes. Membership in a particular ensemble (see below: ensemble-wise loss) is also encoded. The four encodings are concatenated together to form the position embedding $pos(i) = [e_{\text{left}}; e_{\text{post.}}; e_{\text{inf}}; e_{\text{ensemble}}]$, where $e$ is given using a sinusoidal position encoding that represents a scalar coordinate as a unique combination of sines (Vaswani et al., 2017).

The core of PopT consists of a transformer encoder stack (see Appendix A: Architectures). The output of the PopT are spatial-contextual embeddings of the channels $Y = \{y_i\}$ and an embedding of the CLS token $y_{cls}$. During pretraining, the PopT additionally is equipped with a linear head for the [CLS] token output and separate linear heads for all other individual token outputs. These produce the scalars $\tilde{y}_{cls}$ and $\tilde{y}_i$ respectively, which are used in the pretraining objective (Figure 1b and c).

**Self-supervised loss** Our loss has two discriminative components: (1) *ensemble-wise* — the model determines if activity from two ensembles occurred consecutively, requiring an effective brain state representation at the ensemble-level, (2) *channel-wise* — the model identifies outlier channels swapped with a different timepoint's activity, requiring sensitivity to surrounding channel context.

A key aspect of our method is the fact that our objective is discriminative, rather than reconstructive, as is often the case in self-supervision (Liu et al., 2021; Wang et al., 2022). In practice, the temporal embeddings often have low effective dimension (see Wang et al. (2022)), and reconstruction rewards the model for overfitting to "filler" dimensions in the feature vector (Section 5).

**Pretraining** In *ensemble-wise discrimination* (fig. 1b), two different subsets of channels $S_A, S_B \subset C$ are chosen with the condition that they be disjoint $S_A \cap S_B = \emptyset$. During pretraining, the model receives the activities from these channels at separate times $X_A^t = \{x_i^t \mid i \in S_A\}$ and $X_B^{t'} = \{x_i^{t'} \mid i \in S_B\}$. The objective of the task is then to determine whether these states $X_A^t$ and $X_B^{t'}$ have occurred consecutively in time ($|t - t'| = 500ms$) or are separated by some further, randomly selected interval. Given the output of the classification head, the loss function $\mathcal{L}_N$ is the binary cross-entropy (BCE). We select disjoint subsets for ensemble-wise discrimination to prevent the model from solving tasks through trivial copying. We also randomly vary $|S|$ during sampling to ensure the model handles ensembles of different sizes.

In *channel-wise discrimination* (fig. 1c), the model must determine whether a channel's activity has been swapped with activity from a random time. Precisely, activity from each channel $i$ is drawn from a time $t_i = t$ for all channels. Then, 10% of the channels are randomly selected to have their activity replaced with activity from a randomly selected channel at a random point in time $t_i \neq t$. For each of the token outputs of PopT, the channel-wise loss function $\mathcal{L}_C$ is the BCE. Our complete objective function is $\mathcal{L} = \mathcal{L}_N + \mathcal{L}_C$. A detailed formulation of the objective is given in Appendix A.

**Fine-tuning** In fine-tuning, given the [CLS] token, which is a $d$-dimensional vector, the PopT produces the intermediate representation, $\tilde{y}_{cls} \in \mathbb{R}^d$, which is passed through a single layer linear to produce a scalar prediction $\hat{y}_{cls} \in \mathbb{R}$. BCE loss is used for our binary decoding tasks (Section 4).

## 4 EXPERIMENT SETUP

**Data** We use two types of neural time-series data: intracranial and scalp electroencepholography (iEEG and EEG). iEEG probes are surgically implanted within the 3D brain volume and record local electric signals from the brain at very high temporal resolution and spatial precision. EEG electrodes lie on the scalp, and record electric signals that are smeared by the skull, which results

in low temporal and spatial resolution. EEG montages typically tile the whole scalp, while iEEG electrodes are often only inserted in a comparatively smaller number of locations. These cover two resolution extremes of neural time-series data modalities.

*iEEG:* We use the publicly available subject data from Wang et al. (2024). Data was collected from 10 subjects (total 1,688 electrodes, with a mean of 167 electrodes per subject) who watched 26 movies (19 for pretraining, 7 for downstream decoding) while intracranial probes recorded their brain activity. To test decoding with arbitrary ensemble sizes, we select subsets of electrodes based on their individual linear task decodability, with the smallest subsets containing the electrodes with highest decodability. We follow the trialization and data preprocessing practices used in Wang et al. (2022).

*EEG:* We use the Temple University Hospital EEG and Abnormal datasets, TUEG and TUAB (Obeid & Picone, 2016), for pretraining and task data respectively. We remove all task subjects from the pretraining set and follow the data preprocessing practices in Yang et al. (2024); Jiang et al. (2024).

**Decoding Tasks** We evaluate on 5 different classification tasks: 4 auditory-linguistic tasks used in the evaluation of Wang et al. (2022) and 1 widely evaluated abnormal EEG detection task from Obeid & Picone (2016). Of the auditory-linguistic tasks, two of the tasks are audio focused: determining whether a word is spoken with a high or low pitch and determining whether a word is spoken loudly or softly. And two of the tasks have a more linguistic focus: determining whether the beginning of a sentence is occurring or determining whether any speech at all is occurring. The TUAB abnormal EEG detection task is a binary classification of pathological or normal EEG recording.

**Baselines** For controlled baselines, we concatenate the single-channel temporal embeddings and train a linear (Linear) or non-linear (Deep NN) aggregator on the decoding task. These enable us to directly assess how much PopT improves upon existing aggregation approaches (Ghosal & Abbasi-Asl, 2021). These approaches cannot be pretrained across subjects due to the changing meaning and quantity of inputs. To test the effectiveness of pretraining, we also compare against a non-pretrained PopT.

**Methods compared** For the iEEG experiments, we also compare against Brant (Zhang et al., 2024), which is an end-to-end iEEG encoder. We take the fully pretrained Brant model, and fine-tune on our iEEG tasks combining channels with linear aggregation. For the EEG experiments, we compare against reported BIOT (Yang et al., 2024) and LaBraM (Jiang et al., 2024) results. They also train both temporal and spatial encoders together in contrast to our modular approach.

**Temporal encoders** To test the generalizability of our approach, we train with a variety of temporal encoders: BrainBERT (Wang et al., 2022), which is designed for iEEG data, TOTEM (Talukder et al., 2024) which learns a tokenization of the input, Chronos (Ansari et al., 2024) which is a large general time-series encoder, and TS2Vec (Yue et al., 2022) which has a hierarchical convolutional architecture. Hidden dimensions of these encoders vary from 64 to 768. More details are in Appendix A.

| Model | Pitch | Volume | Sent. Onset | Speech/Non-speech |
|---|---|---|---|---|
| BrainBERT: | | | | |
|     Linear Agg. | $0.59 \pm 0.03$ | $0.66 \pm 0.03$ | $0.70 \pm 0.04$ | $0.71 \pm 0.04$ |
|     Deep NN Agg. | $0.56 \pm 0.03$ | $0.64 \pm 0.04$ | $0.71 \pm 0.03$ | $0.70 \pm 0.04$ |
|     Non-pretrained PopT | $0.53 \pm 0.02$ | $0.61 \pm 0.05$ | $0.74 \pm 0.04$ | $0.70 \pm 0.03$ |
|     **Pretrained PopT** | $\mathbf{0.74 \pm 0.03}^*$ | $\mathbf{0.87 \pm 0.03}^*$ | $\mathbf{0.90 \pm 0.01}^*$ | $\mathbf{0.93 \pm 0.02}^*$ |
| TOTEM: | | | | |
|     Linear Agg. | $0.57 \pm 0.02$ | $0.67 \pm 0.02$ | $0.80 \pm 0.03$ | $0.77 \pm 0.05$ |
|     Deep NN Agg. | $0.58 \pm 0.02$ | $0.70 \pm 0.02$ | $0.80 \pm 0.03$ | $0.77 \pm 0.05$ |
|     Non-pretrained PopT | $0.53 \pm 0.01$ | $0.62 \pm 0.02$ | $0.80 \pm 0.03$ | $0.77 \pm 0.05$ |
|     **Pretrained PopT** | $\mathbf{0.64 \pm 0.03}$ | $\mathbf{0.79 \pm 0.02}^*$ | $\mathbf{0.90 \pm 0.02}^*$ | $\mathbf{0.88 \pm 0.05}^*$ |
| End-to-end: | | | | |
|     Brant (Zhang et al., 2024) | $\mathbf{0.61 \pm 0.03}$ | $\mathbf{0.74 \pm 0.03}$ | $\mathbf{0.80 \pm 0.04}$ | $\mathbf{0.80 \pm 0.03}$ |

Table 1: **Pretraining PopT is critical to downstream decoding performance (iEEG data).** We test on a variety of audio-linguistic decoding tasks (see Section 4) with 90 channels as input. The temporal encoder used for aggregation in sections 1 and 2 are denoted in the section header. We also evaluate against an end-to-end pretrained iEEG model in section 3. Shown are the ROC-AUC mean and standard error across subjects. Best per section are bolded. Asterisks $^*$ indicate that the bolded model is significantly better than the second-place model ($p < 0.05$, Wilcoxon rank-sum).

| Model | Balanced Accuracy | ROC AUC |
|---|---|---|
| **Chronos:** | | |
| Linear Agg. | $0.7754 \pm 0.0008$ | $0.8563 \pm 0.0003$ |
| Deep NN Agg. | $0.7881 \pm 0.0057$ | $0.8678 \pm 0.0049$ |
| Non-pretrained PopT | $0.7763 \pm 0.0047$ | $0.8631 \pm 0.0016$ |
| **Pretrained PopT** | $\mathbf{0.7963 \pm 0.0021^*}$ | $\mathbf{0.8822 \pm 0.0013^*}$ |
| **TS2Vec:** | | |
| Linear Agg. | $0.7554 \pm 0.0005$ | $0.8549 \pm 0.0011$ |
| Deep NN Agg. | $0.7835 \pm 0.0039$ | $0.8723 \pm 0.0029$ |
| Non-pretrained PopT | $0.7896 \pm 0.0037$ | $0.8782 \pm 0.0018$ |
| **Pretrained PopT** | $\mathbf{0.8060 \pm 0.0025^*}$ | $\mathbf{0.8883 \pm 0.0008^*}$ |
| **End-to-end:** | | |
| BIOT (Yang et al., 2024) | $0.7959 \pm 0.0057$ | $0.8815 \pm 0.0043$ |
| LaBraM (Jiang et al., 2024) | $\mathbf{0.8258 \pm 0.0011}$ | $\mathbf{0.9162 \pm 0.0016}$ |

Table 2: **Pretraining PopT is critical to downstream decoding performance (EEG data).** We test on an abnormal EEG detection task (see Section 4) with 21 channels as input. The temporal encoder used for aggregation in sections 1 and 2 are denoted in the section header. We also evaluate against end-to-end pretrained EEG models in section 3 (values from the original works). Shown are mean and standard deviation across 5 random seeds. Best per section are bolded. Asterisks * indicate that the bolded model is significantly better than the second-place model ($p < 0.05$, Wilcoxon rank-sum).

## 5 RESULTS

**Decoding performance** We find that using a pretrained PopT significantly benefits downstream decoding compared to baseline channel aggregation techniques across tasks, data modalities, and temporal encoding models (Tables 1 and 2 and Figure 2). To test our method's ability to handle multiple types of channel encodings, we applied our framework to 4 different channel encoders: (1) an iEEG-specific temporal encoder: BrainBERT (Wang et al., 2022), (2) a general tokenization-based time-series encoder: TOTEM (Talukder et al., 2024), (3) a pretrained general time-series encoder: Chronos (Ansari et al., 2024), and a general convolution-based time-series encoder: TS2Vec (Yue et al., 2022). We see significant improvements in performance with the pretrained PopT in all cases when comparing with baseline aggregation approaches (Figure 2). Additionally, the pretrained PopT scales well with increasing ensemble sizes (Figure 3), a challenging task for the baseline aggregation approaches due to limited downstream task data and increasing input size.

We also find that PopT can achieve competitive performance against pretrained end-to-end models, such as Brant (Zhang et al., 2024) for iEEG, and BIOT (Yang et al., 2024) and LaBraM (Jiang et al., 2024) for EEG (Tables 1 and 2). For instance, PopT outperforms Brant (Zhang et al., 2024) in decoding iEEG data with our pretrained PopT + BrainBERT combination, likely due to PopT's ability to leverage spatial relationships. Whereas Brant leaves the channel aggregation problem open.

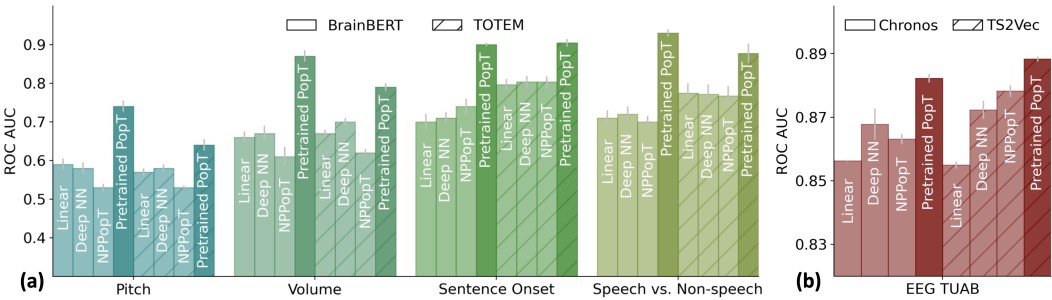

Figure 2: **Compared to common aggregation approaches, pretrained PopT consistently yields better downstream decoding across tasks, data modalities, and temporal embedding types.** NPopT = Non-pretrained PopT. (a) performance on four audio-linguistic iEEG tasks with 90 electrodes. Grey bars denote standard error across subjects. (b) performance on an abnormal detection EEG task with 21 electrodes. Grey bars denote standard deviation across 5 random seeds.

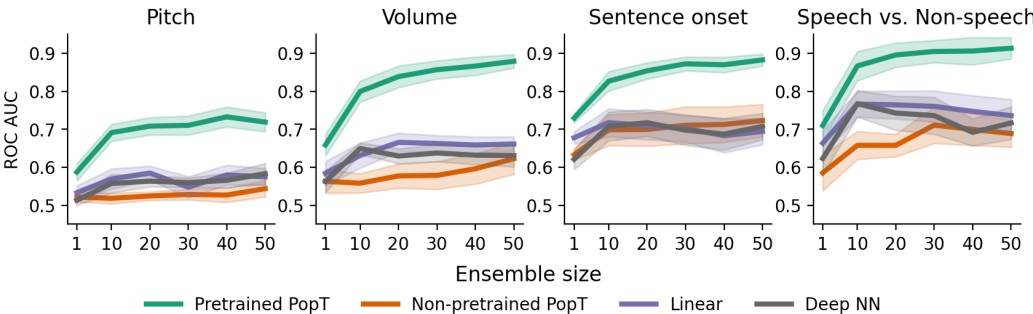

Figure 3: **Pretrained PopT downstream performance scales better with ensemble size.** Increasing channel ensemble size from 1 to 50 (x-axis), we see pretrained PopT (green) decoding performance (y-axis) not only beat non-pretrained approaches (orange, purple, grey), but also continually improve more with increasing channel count. Shaded bands show the standard error across subjects.

PopT is competitive with recent end-to-end trained EEG models (Yang et al., 2024; Jiang et al., 2024) on the EEG TUAB abnormal detection task. This is impressive, since models such as LaBraM were specifically developed for this application, whereas PopT was trained on top of generic time-series embeddings. We find that PopT can offer an efficient and competitive alternative to large end-to-end models for these decoding tasks, due to the effectiveness of our pretraining task for learning spatial and functional relationships between channel input embeddings.

To verify that the weights of the pretrained PopT capture neural processing well even without fine-tuning, we also train a linear-encoder on top of the frozen PopT `[CLS]` token and find the same trends (Figure 17). This point in particular is important in building confidence in the results of our interpretability studies (Section 6), in which we use the frozen pretrained weights to analyze connectivity. For the remaining analyses described below, we use a PopT with BrainBERT inputs.

**Sample and compute efficiency** Our PopT learns spatial relationships between channels, in a way that makes downstream supervised learning more data and compute efficient (Figure 4 and Figure 5). Compared to the non-pretrained baseline models, fine-tuning the pretrained PopT can achieve the same decoding performance as other aggregation techniques with an order of magnitude fewer samples. The pretrained PopT surpasses the performance achieved by all other aggregation techniques by 500 samples out of the full dataset (roughly 5-10k examples depending on subject and task) (Figure 4). The pretrained PopT also converges at a low number of steps. This greatly contrasts with the non-pretrained PopT. The Linear and Deep NN baselines can be similarly compute efficient, but occasionally may require 2k or more steps (Figure 5), as in the case of Speech vs. Non-speech.

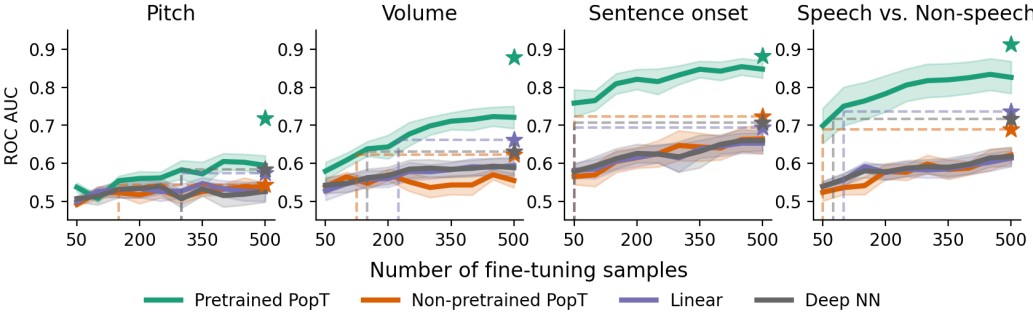

Figure 4: **Pretrained PopT is more sample efficient when fine-tuning.** Varying the number of samples available to each model at train time (x-axis), we see that the pretrained PopT is highly sample efficient, requiring only a fraction of samples (fewer than 500 samples out of 5-10k of the full dataset) to reach the full performance level of baseline aggregation approaches (dashed lines). Bands show standard error across test subjects. Stars indicate performance with full fine-tuning dataset.

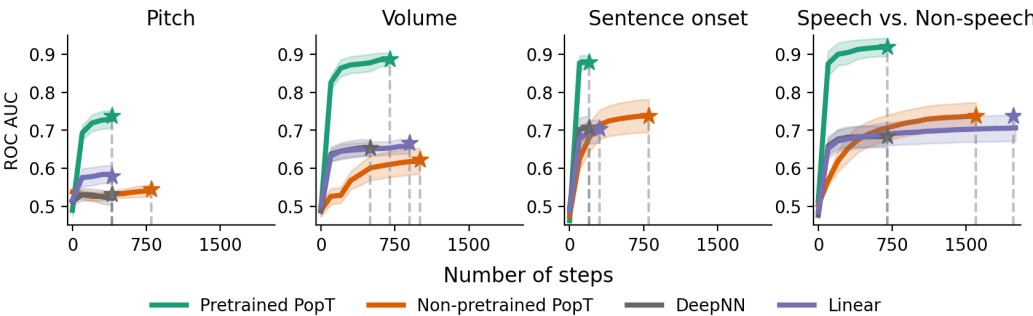

Figure 5: **Pretrained PopT is consistently compute efficient when fine-tuning.** Number of steps required for each model to reach final performance during fine-tuning (dashed lines). The pretrained PopT consistently requires fewer than 750 steps (each step is an update on a batch size of 256) to converge. Bands show standard error across subjects. Stars indicate fully trained performance.

**Generalizability** To test if our pretrained weights will be useful for subjects not seen during training, we conduct a hold-one-out analysis. We pretrain a model using all subjects except for one, and then fine-tune and evaluate on the held-out subject. We find that missing a subject from pretraining does not significantly affect the downstream results (Figure 6). This raises our confidence that the pretrained weights will be useful for unseen subjects and for researchers using new data.

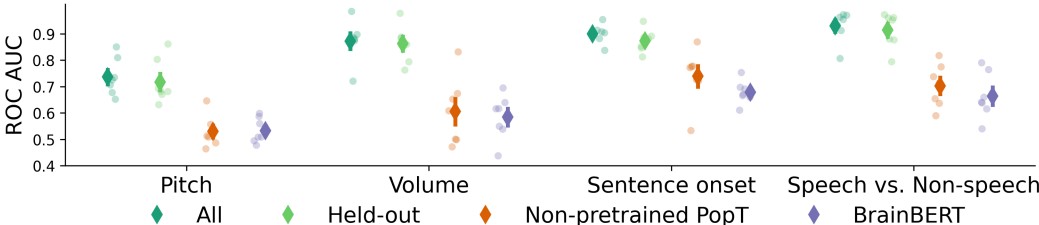

Figure 6: **Gains in decoding performance are available to new subjects.** A minimal decrease in downstream decoding performance is found if the subject is held-out from pretraining (Held-out vs All). Results are cross-validated across all test subjects. For BrainBERT, we report performance on the channel with the best linear-decodability. Markers show mean and standard error across subjects.

**Scaling with amount of pretraining data** To investigate the effect of scaling pretraining data on our model, we pretrain versions of PopT using only 2%, 5%, 10%, 25%, and 50% of our data. Evaluation is performed on all test-subjects. We find a general improvement in downstream decoding when we increase the amount of pretraining data available across all our downstream decoding tasks (Figure 7), suggesting decoding improvements as we double our pretraining dataset. We expect that the framework could benefit from more diverse data, as a slight plataeuing effect is seen with the current pretraining dataset.

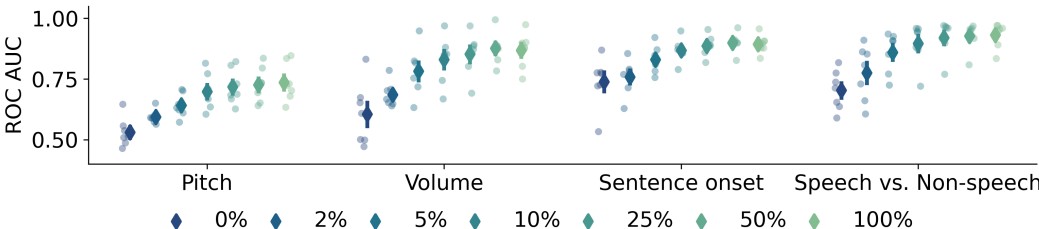

Figure 7: **Pretraining with more data leads to better downstream performance.** We pretrain PopT with different percentages of our full pretraining dataset (colors) and test on our decoding tasks (x-axis). Markers show mean and standard error across test subjects.

**Ablation of loss components and position information** An ablation study confirms that both ensemble-wise and channel-wise losses contribute to downstream performance (Table 3). Furthermore, including the 3D position information for each channel and using discriminative losses are critical. Removing our positional encoding during pretraining and fine-tuning drops the performance significantly. Attempting to only use an L1 reconstruction term for our pretraining objective results in poorer performance. Our discriminative loss requires the model to understand the embeddings in terms of how they can be distinguished from one another, which leads the model to extract representations that are more beneficial for decoding.

|  | Pitch | Volume | Sent. Onset | Speech/Non-speech |
|---|---|---|---|---|
| PopT | $\mathbf{0.74 \pm 0.03}$ | $\mathbf{0.87 \pm 0.03}$ | $\mathbf{0.90 \pm 0.01}$ | $\mathbf{0.93 \pm 0.02}$ |
| PopT w/o channel-wise loss | $0.71 \pm 0.03$ | $0.86 \pm 0.03$ | $0.88 \pm 0.02$ | $0.92 \pm 0.02$ |
| PopT w/o ensemble-wise loss | $0.70 \pm 0.03$ | $0.87 \pm 0.02$ | $0.87 \pm 0.01$ | $0.92 \pm 0.02$ |
| PopT w/o position encoding | $0.62 \pm 0.07^{\vee}$ | $0.76 \pm 0.07^{\vee}$ | $0.81 \pm 0.09^{\vee}$ | $0.83 \pm 0.10^{\vee}$ |
| PopT reconstruction loss only | $0.56 \pm 0.04^{\vee}$ | $0.65 \pm 0.08^{\vee}$ | $0.73 \pm 0.10^{\vee}$ | $0.74 \pm 0.10^{\vee}$ |

Table 3: **PopT ablation study.** We individually ablate our losses and positional encodings during pretraining then decode on the resulting models. Shown are ROC-AUC mean and standard error across subjects evaluated at 90 electrodes. The best performing model across all decoding tasks uses all of our proposed components. Here, $\vee$ denotes ablations which are significantly worse than the full model ($p < 0.05$, Dunnett's test).

## 6 INTERPRETING LEARNED WEIGHTS

**Connectivity** Traditional neuroscience analyses typically use cross-correlation as a measure of region connectivity (Wang et al., 2021). Our PopT allows for an alternative method of determining connectivity, based on the degree to which channels are sensitive to each other's context. In this method, each channel is masked in turn, and then model performance on the pretraining channel-wise objective for the remaining unmasked channels is measured. We use the degradation in performance as a measure of connectivity. We can construct plots (Figure 8) that recapitulate the strongest connectivity of the cross-correlation maps. Note that while some approaches for modelling brain activity explicitly build this into their architecture (Cai et al., 2023), we recover these connections purely as a result of our self-supervised learning. Additional method details available in Appendix G.

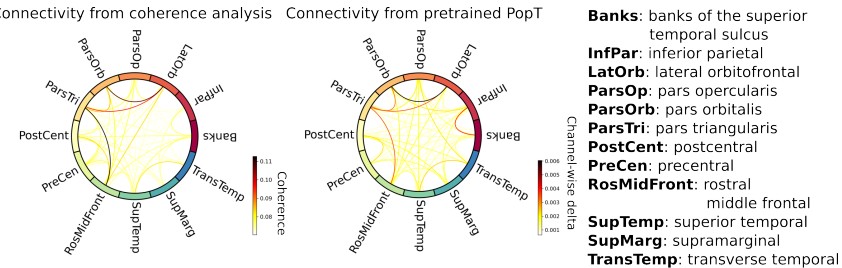

Figure 8: **Probing the pretrained model for inter-channel connectivity.** Traditionally, connectivity analysis between regions is done by computing the coherence between electrode activity (left). We propose an alternative analysis purely based on the contextual sensitivity learned during pretraining. Briefly, we select an electrode, mask out its activity, and then measure the degradation in the channel-wise objective function for the remaining electrodes. Plotting the values of this delta (right) recovers the main points of connectivity. Plots for all test subjects can be seen in Appendix I.

**Candidate functional brain regions from attention weights** After fine-tuning our weights on a decoding task, we can examine the attention weights of the `[CLS]` output for candidate functional brain regions. We obtain a normalized Scaled Attention Weight metric across all subjects to analyze candidate functional brain regions across sparsely sampled subject datasets (Figure 9). The Scaled Attention Weight is computed from raw attention weights at the `[CLS]` token passed through the attention rollout algorithm (Abnar & Zuidema, 2020). The resulting weights from each channel are then grouped by brain region according to the Desikan-Killiany-Tourville (DKT) region atlas (Klein & Tourville, 2012). A full description of the method is available in Appendix G.

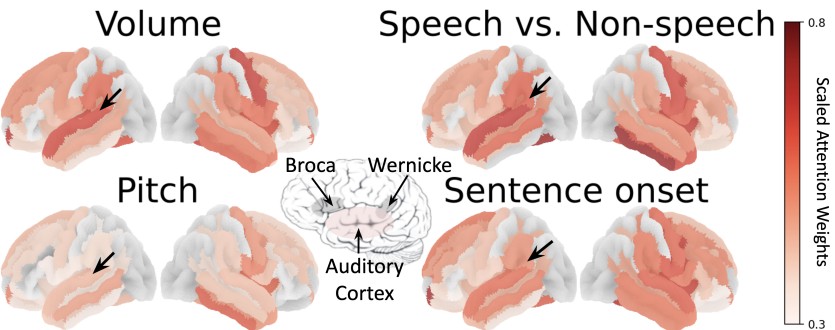

Figure 9: **Attention weights from fine-tuned PopT identify candidate functional brain regions** Candidate functional maps can be read from attention weights of a PopT fine-tuned on our decoding tasks. For the Speech vs Non-speech and Sentence onset tasks, note the weight placed on regions near Wernicke's area (black arrows). In all tasks, the auditory cortex is attended to. Center brain figure highlight regions related to auditory-linguistic processing; figure credit: (aph, 2017)).

The resulting weights reveal expected functional brain regions related to the tasks decoded (Figure 9), with low-level auditory tasks highlighting primary auditory cortex and higher-level language distinction tasks highlighting language-specific areas. Given the training PopT undergoes, these weights provide a technique for discovering candidate regions (see Appendix H for quantitative comparison).

## 7 DISCUSSION

We presented a self-supervised scheme for learning effective joint representations of neural activity from temporal embeddings. Our approach improves decoding and reduces the samples required to learn downstream tasks, which is especially critical for neural data modalities given subject constraints. A key aspect of our approach is the fact that we focus on spatial aggregation of existing channel embeddings, rather than training a large end-to-end model. By decoupling temporal and spatial feature extraction, we are able to leverage existing temporal embeddings to learn spatiotemporal representations efficiently and with a smaller number of parameters. This makes our model available for use in low compute-resource settings. Furthermore, this separation of considerations opens up the possibility for future independent improvement in temporal modeling, whether that be from a domain specific model or a more general time-series encoder. The generality of this approach allowed us to train on two very different neural modalities: scalp EEG and invasive iEEG. Our success in these domains suggest that this approach could even be extended to settings outside of neuroscience that also contend with sparsely and variably distributed time-series data channels, as is often the case with geophysical or climate data.

**Limitations and Future Work** We proposed a strategy for aggregating signals, provided that meaningful spatial coordinates are available, but it remains to be seen how to extend this approach to settings without such coordinates. Electrode layouts are highly variable, so it is important that some notion of positional encoding be given. Future work could experiment with automatic functional identification for each channel, such as that explored in neural spiking data (Azabou et al., 2024), but it is currently unclear how to do so with neural recordings that have lower SNR.

## 8 CONCLUSION

We introduced a pretraining method for learning representations of arbitrary ensembles of intracranial electrodes. We showed that our pretraining produced considerable improvements in downstream decoding and efficiency, that would not have been possible without the knowledge of spatial relationships learned during the self-supervised pretraining stage. These benefits were found across data modalities, decoding tasks, and temporal encoders used, speaking to the generality of our approach. We further showed that this scheme produces interpretable weights from which connectivity maps and candidate functional brain regions can be read. Finally, we release the pretrained weights for our PopT with BrainBERT inputs as well as our code for pretraining with any temporal embedding.

## 9 Acknowledgements

We would like to thank Xuefei (Julie) Wang and Kejun (Amy) Li for helpful comments and discussion.

We would like to thank our funding sources: Caltech Chen Institute, Caltech Carver Mead New Adventures Fund, Center for Brains, Minds, and Machines, NSF STC award CCF-1231216, the NSF award 2124052, the MIT CSAIL Machine Learning Applications Initiative, the MIT-IBM Watson AI Lab, the CBMM-Siemens Graduate Fellowship, the DARPA Mathematics for the DIscovery of ALgorithms and Architectures (DIAL) program, the DARPA Knowledge Management at Scale and Speed (KMASS) program, the DARPA Machine Common Sense (MCS) program, the Air Force Office of Scientific Research (AFOSR) under award number FA9550-21-1-0399, and the United States Air Force Research Laboratory and the Department of the Air Force Artificial Intelligence Accelerator under Cooperative Agreement Number FA8750-19-2-1000. The views and conclusions contained in this document are those of the authors and should not be interpreted as representing the official policies, either expressed or implied, of the Department of the Air Force or the U.S. Government. The U.S. Government is authorized to reproduce and distribute reprints for Government purposes notwithstanding any copyright notation herein.

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

# A  ARCHITECTURES AND TRAINING

**Pretrained PopT** The core Population Transformer consists of a transformer encoder stack with 6 layers, 8 heads. All layers in the encoder stack are set with the following parameters: $d_h = 512$, $H = 8$, and $p_{\text{dropout}} = 0.1$. We pretrain the PopT model with the LAMB optimizer (You et al., 2019) ($lr = 5e-4$), with a batch size of $n_{\text{batch}} = 256$, and train/val/test split of 0.89, 0.01, 0.10 of the data. We pretrain for 500,000 steps, and record the validation performance every 1,000 steps. Downstream evaluation takes place on the weights with the best validation performance. We use the intermediate representation at the $\texttt{[CLS]}$ token $d_h = 512$ and put a linear layer that outputs to $d_{out} = 1$ for fine-tuning on downstream tasks. These parameters for pretraining were the same for any PopT that needed to be pretrained (across temporal embeddings, hold-one-out subject, ablation studies).

**Pretraining task: Ensemble-wise pretraining** Two different subsets of channels $S_A, S_B \subset C$ are chosen with the condition that they be disjoint $S_A \cap S_B = \emptyset$. During pretraining, the model receives the activities from these channels at separate times $X_A = \{x_i^t \mid i \in S_A\}$ and $X_B = \{x_i^{t'} \mid i \in S_B\}$. The sets $X_A$ and $X_B$ can be written as an $S_A \times d$ matrix and $S_B \times d$ matrix respectively. The PopT receives these matrices as input, along with the token $\texttt{[CLS]}$. The objective of the task is then to determine whether these states $X_A$ and $X_B$ have occurred consecutively in time or are separated by some further, randomly selected interval. The PopT produces outputs for all inputs, including the classification head, $\tilde{y}_{cls} \in \mathbb{R}^d$. Then, $\tilde{y}_{cls}$ passes through a linear layer to produce a scalar $\hat{y}_{cls} \in \mathbb{R}$. The objective function is the BCE between this prediction and the label $y_{cls}^*$: $\mathcal{L}_N = \frac{1}{|S_A|+|S_B|} \sum_i y_{cls}^* \log(p(\hat{y}_{cls})) + (1 - y_{cls}^*) \log(p(\hat{y}_{cls}))$, where $y_{cls}^* = \mathbf{1}(|t - t'| < 500ms)$

**Pretraining task: Channel-wise pretraining** The token level objective is to determine whether a channels activity has been swapped with activity from a random time. Precisely, activity from each channel $i$ is drawn from a time $t_i$. All channels are drawn from the same time $t_i = t$, and then 10% of the channels are randomly selected to have their activity replaced with activity from a randomly selected channel, taken from a random point in time $t_i \neq t$. Then, the channel-wise outputs, $\tilde{y}_i \in \mathbb{R}^d$, of the Population Transformer are passed through a linear layer to obtain scalar predictions $\hat{y}_i$. The objective function is the BCE between these predictions and the labels $y_i^*$: $\mathcal{L}_C = \frac{1}{|S_A|+|S_B|} \sum_i y_i^* \log(p(\hat{y})) + (1 - y_i^*) \log(p(\hat{y}_i))$ where $y_i^* = \mathbf{1}(t_i \neq t)$. Then, the complete pretraining objective is $\mathcal{L} = \mathcal{L}_C + \mathcal{L}_N$

**Non-pretrained PopT** The architecture for the non-pretrained PopT is the same as the pretrained PopT (above). However, no pretraining is done, and the weights are randomly initialized with the default initializations.

**Linear** The linear baseline consists of a single linear layer that outputs to $d_{out} = 1$. The inputs are flattened and concatenated BrainBERT embeddings $d_{emb} = 756$, TOTEM embeddings $d_{emb} = 64$, Chronos embeddings $d_{emb} = 512$, or TS2Vec embeddings $d_{emb} = 320$ from a subset of channels $S \subset C$. Thus, the full input dimension is $d_{input} = d_{emb} * |S|$.

**Deep NN** The inputs are the same as above, but the decoding network now consists of 5 stacked linear layers, each with $d_h = 512$ and a GeLU activation.

**Downstream Training** For PopT models, we train with these parameters: AdamW optimizer (Loshchilov & Hutter, 2017), $lr = 5e^{-4}$ where transformer weights are scaled down by a factor of 10 ($lr_t = 5e^{-5}$), $n_{batch} = 128$, a Ramp Up scheduler (ildoonet, 2024) with warmup 0.025 and Step LR gamma 0.95, reducing 100 times within the 2000 total steps that we train for. For Linear and DeepNN models, we train with these parameters: AdamW optimizer (Loshchilov & Hutter, 2017), $lr = 1e^{-3}$, $n_{batch} = 256$, a Ramp Up scheduler (ildoonet, 2024) with warmup 0.025 and Step LR gamma 0.95, reducing 100 times within the 17,000 total steps we train for. For all downstream decoding, we use a fixed train/val/test split of 0.8, 0.1, 0.1 of the data. For non-BrainBERT models, we provide configuration files for downstream decoding parameters in our codebase.

**Compute Resources** To run all our experiments (data processing, pretraining, evaluations, interpretability), one only needs 1 NVIDIA Titan RTXs (24GB GPU RAM). Pretraining PopT takes 2 days on 1 GPU. Our downstream evaluations take a few minutes to run each. For the purposes of data processing and gathering all the results in the paper, we parallelized the experiments on 8 GPUs.

# B    MODEL AND COMPUTE REQUIREMENTS

|  | e5 | e50 | e90 |
|---|---|---|---|
| PopT |  | 20M |  |
| Deep NN | 3M | 20M | 36M |
| Linear | 3.8k | 38k | 69k |
| Brant (Zhang et al., 2024) |  | 500M |  |
| LaBraM (Jiang et al., 2024) |  | 350M |  |
| BIOT (Yang et al., 2024) |  | 3M |  |

Table 4: **Parameter counts**. Since PopT takes existing temporal embeddings as input, the number of trainable parameters is an order of magnitude less than some recent end-to-end approaches.

|  | # GPUs | GPU type | Time to train | TFLOPS |
|---|---|---|---|---|
| PopT | 1 | NVIDIA TITAN RTX (24GB) | 2 days | 2.1M |
| Brant (Zhang et al., 2024) | 4 | NVIDIA Tesla A100 (80G) | 2.8 days | 18.8M |
| LaBraM (Jiang et al., 2024) | 8 | NVIDIA Tesla A800 (40G) | – | – |
| BIOT (Yang et al., 2024) | 8 | NVIDIA RTX A6000 (48G) | – | – |

Table 5: **Pretraining compute requirements** Based on published train times (none were given for LaBraM and BIOT) it is evident that PopT has smaller hardware and shorter training time requirements.

# C    DECODING TASKS

We follow the same task specification as in Wang et al. (2022), with the modification that the pitch and volume examples are determined by percentile (see below) rather than standard deviation in order to obtain balanced classes.

**Pitch** The PopT receives an interval of activity and must determine if it corresponds with a high or low pitch word being spoken. For the duration of a given word, pitch was extracted using Librosa's `piptrack` function over a Mel-spectrogram (sampling rate 48,000 Hz, FFT window length of 2048, hop length of 512, and 128 mel filters). For this task, for a given session, positive examples consist of words in the top-quartile of mean pitch and negative examples are the words in the bottom quartiles.

**Volume** The volume of a given word was computed as the average intensity of root-mean-square (RMS) (`rms` function, frame and hop lengths 2048 and 512 respectively). As before, positive examples are the words in the top-quartile of volume and negative examples are those in the bottom quartiles.

**Sentence onset** Negative examples are intervals of activity from 1s periods during which no speech is occurring in the movie. Positive examples are intervals of brain activity that correspond with hearing the first word of a sentence.

**Speech vs. Non-speech** Negative examples are as before. Positive examples are intervals of brain activity that correspond with dialogue being spoken in the stimuli movie.

## D    DATASET DETAILS

| Subj. | Age (yrs.) | # Electrodes | Movie | Recording time (hrs) | Held-out |
|---|---|---|---|---|---|
| 1 | 19 | 91 | Thor: Ragnarok | 2.07 | |
| | | | Fantastic Mr. Fox | 1.91 | |
| | | | The Martian | 2.90 | x |
| 2 | 12 | 100 | Venom | 2.60 | |
| | | | Spider-Man: Homecoming | 2.42 | |
| | | | Guardians of the Galaxy | 2.66 | |
| | | | Guardians of the Galaxy 2 | 3.00 | |
| | | | Avengers: Infinity War | 3.73 | |
| | | | Black Panther | 1.85 | |
| | | | Aquaman | 3.52 | x |
| 3 | 18 | 91 | Cars 2 | 1.90 | x |
| | | | Lord of the Rings 1 | 2.94 | |
| | | | Lord of the Rings 2 (extended edition) | 4.06 | |
| 4 | 12 | 152 | Incredibles | 1.31 | |
| | | | Shrek 3 | 1.87 | x |
| | | | Megamind | 1.75 | |
| 5 | 6 | 109 | Fantastic Mr. Fox | 1.54 | |
| 6 | 9 | 135 | Megamind | 0.81 | |
| | | | Toy Story | 1.32 | |
| | | | Coraline | 1.6 | x |
| 7 | 11 | 205 | Cars 2 | 1.67 | x |
| | | | Megamind | 1.77 | |
| 8 | 4.5 | 121 | Sesame Street Episode | 1.41 | |
| 9 | 16 | 72 | Ant Man | 1.0 | |
| 10 | 12 | 173 | Cars 2 | 1.57 | x |
| | | | Spider-Man: Far from Home | 2.33 | |

Table 6: **Subject statistics** Subjects used in BrainBERT training, and held-out downstream evaluation. The number of uncorrupted, electrodes that can be Laplacian re-referenced are shown in the second column. The average amount of recording data per subject is 5.55 (hrs).

## E    HOLD OUT SUBJECT PRETRAINING GENERALIZABILITY

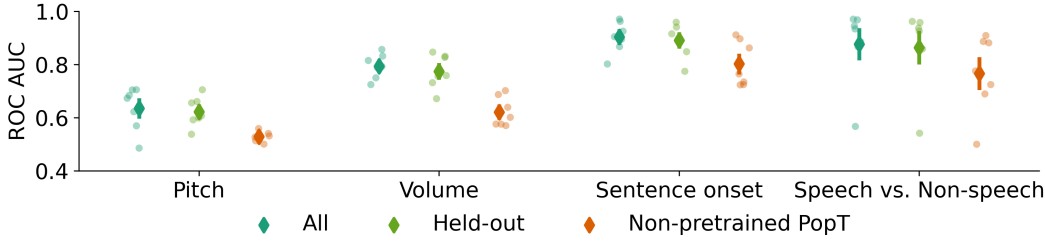

Figure 10: **Gains in decoding performance are available to new subjects even on TOTEM pretrained PopT.** Same experiment as Figure 6 but with TOTEM embedding.

## F    RANDOM ELECTRODE ENSEMBLE PERFORMANCE

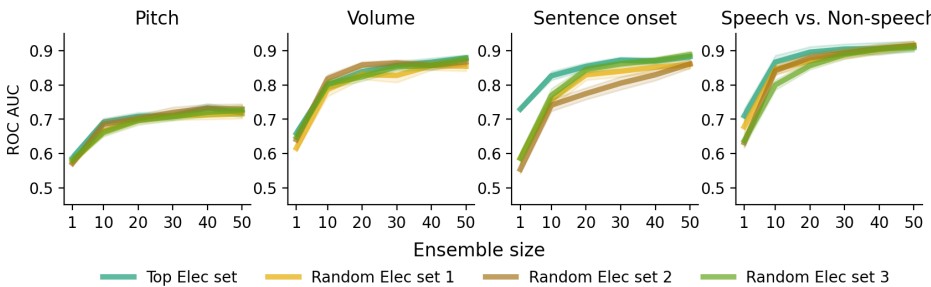

Figure 11: **Downstream decoding performance on random electrode subsets.** To check if our original channel ensemble ordering inflated performance, we perform downstream decoding on 3 randomly generated electrode ensembles. The random electrode ensembles perform roughly similar to our reported values, with the exception of a few low-electrode count ensembles for Sentence Onset. These exceptions may be due to strong decodability of Sentence Onset at specific electrodes. Shaded bands show the standard error across subjects.

## G    INTERPRETATION METHODS

**Connectivity analysis**  We start with a pretrained PopT. To test a particular channel's contribution to connectivity, we omit it from the input (more details in Appendix I). Then, we consider the remaining unmasked channels and ask: how does this change the pretraining channel-wise loss? Recall that this objective is to determine if a channel has had its inputs swapped with random activity. If the change in loss is large, we infer that the masked channel provided important context. Using the magnitude of this delta as a measure for connectivity, we then average across the Desikan-Killiany regions (Desikan et al., 2006) and produce a plot using `mne-connectivity` (Gramfort et al., 2013).

**Scaled Attention Weight**  First, we obtain an attention weight matrix across all trials which includes weights between all tokens. Then, we perform attention rollout (Abnar & Zuidema, 2020) across layers to obtain the contributions of each input channel by the last layer. We take the resulting last layer of rollout weights for all channels, where the target is the `[CLS]` token, normalize within subject, and scale by ROC AUC to obtain the Scaled Attention Weight per channel. Finally, we take the 0.75 percentile score per DKT region (Klein & Tourville, 2012), average across 5 fine-tuning runs, and plot using Nilearn (contributors).

## H    FUNCTIONAL BRAIN REGION COMPARISON

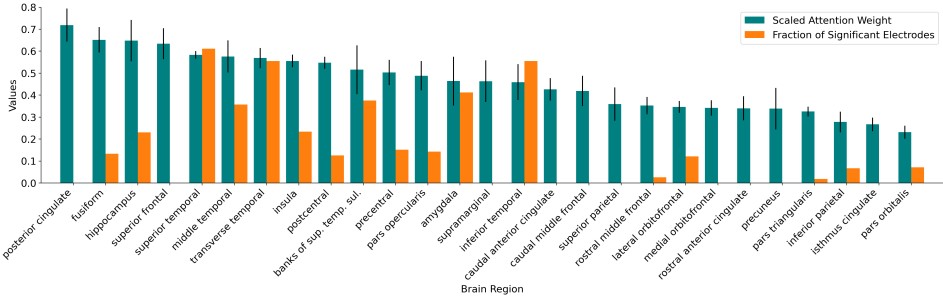

Figure 12: **Comparison of functional maps as identified by our method vs traditional measures.** Scaled Attention Weight vs Fraction of Significant Electrodes per DKT (Klein & Tourville, 2012) region for the Speech vs. Non-speech task. Fraction of significant electrodes from Wang et al. (2024). The Pearson's $r$ correlation coefficient is 0.4 between the two metrics. Error bars are standard deviation across 5 fine-tuning runs.

## I  CONNECTIVITY

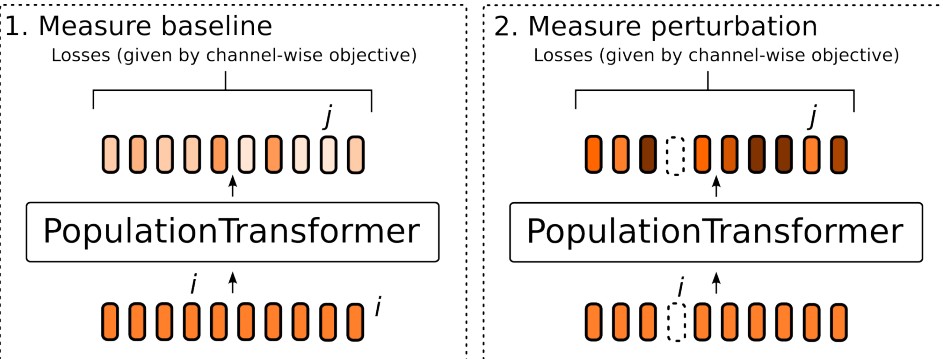

Figure 13: **Schematic of connectivity analysis** To determine the influence of some channel, $i$, on another channel $j$, we first measure the baseline performance of the pretrained PopT on the replace-only objective. Then, we omit $i$ from the input, and measure how the performance on the channel-wise objective is perturbed on $j$. See also Algorithm 1.

---

**Algorithm 1** Connectivity measurement between channels $i$ and $j$

---

**Require:** $j < i, x \in \mathbb{R}^{N_C \times d}$      ▷ $N_C$ is the number of channels, $d$ is the embedding dimension.
    $\hat{y}_{\text{baseline}} \leftarrow P(x)$      ▷ $P$ is a pretrained PopT, $\hat{y}_{\text{baseline}} \in \mathbb{R}^{N_C}$
    **while** $n \le N_{\text{samples}}$ **do**
        $x_{\text{omitted}} \leftarrow \text{Concat}(x[: i], x[i + 1 :])$      ▷ Remove the $i^{th}$ channel from the input
        $\hat{y}_{\text{perturbed}} \leftarrow P(x_{\text{omitted}})$
        $\text{Influence} = |\hat{y}_{\text{baseline}} - \hat{y}_{\text{perturbed}}|$      ▷ How much did the prediction change?
        $\text{AvgConnectivity} \leftarrow \text{AvgConnectivity} + \text{Influence}[j]/n$
    **end while**

---

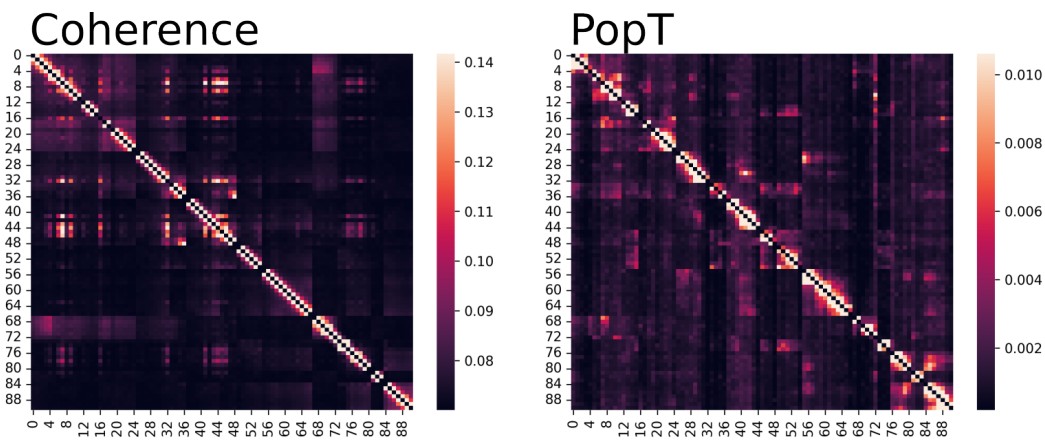

Figure 14: **Electrode level connectivity.** Connectivity between all channels for the same subject shown in Figure 8. Outliers at the 2-percentile are snapped to color map floor and ceiling.

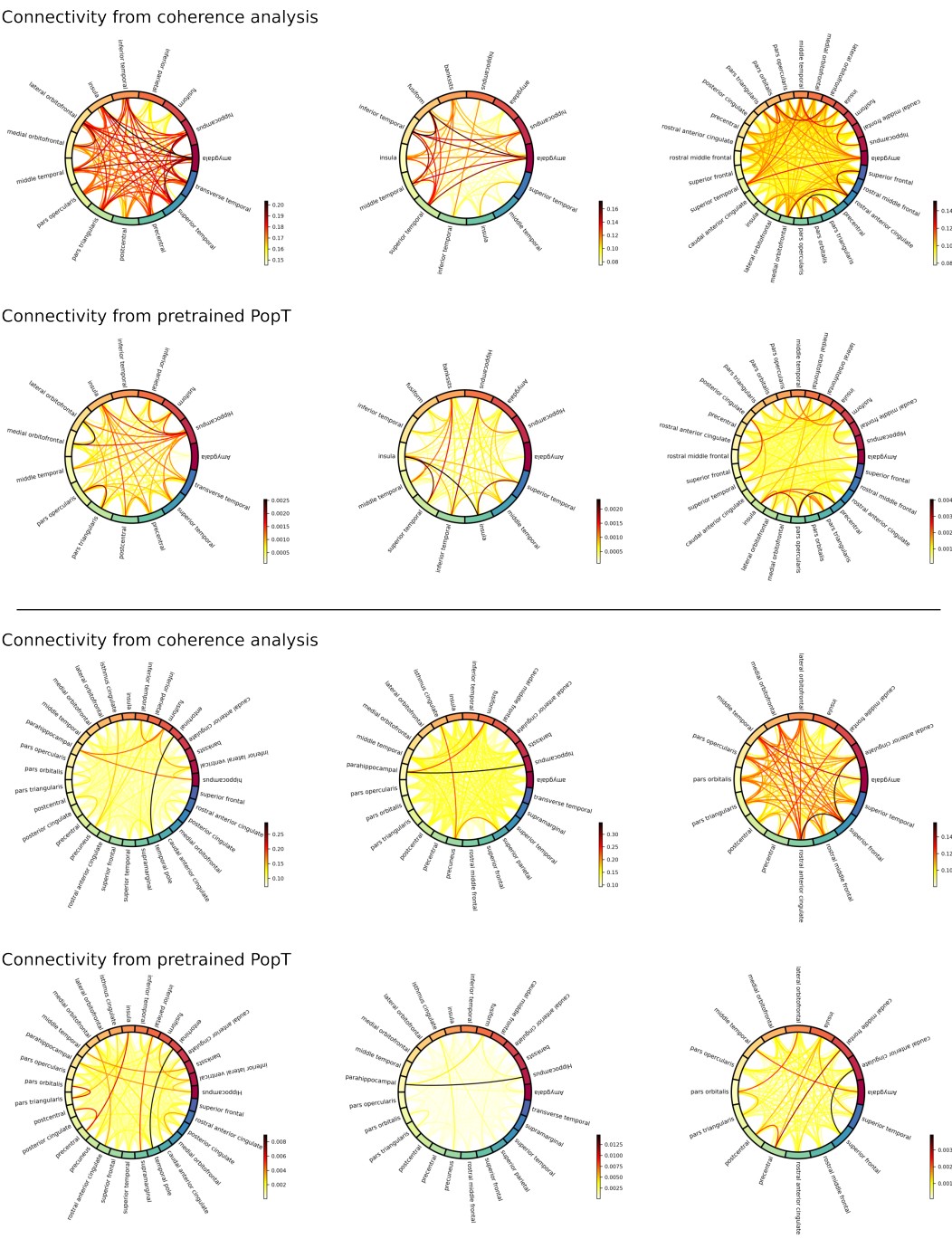

Figure 15: **Region connectivity for test subjects.** Continued from Figure 8; this figures shows the rest of the test subjects. We compare between traditional connectivity analysis performed via coherence (top row in each section) and the analysis based on our PopT pretrained weights (bottom row in each section). We note that our analysis usually recovers the strongest points of connectivity from the traditional analysis. Coherence was computed using scikit-learn's (Pedregosa et al., 2011) `signal.coherence`.

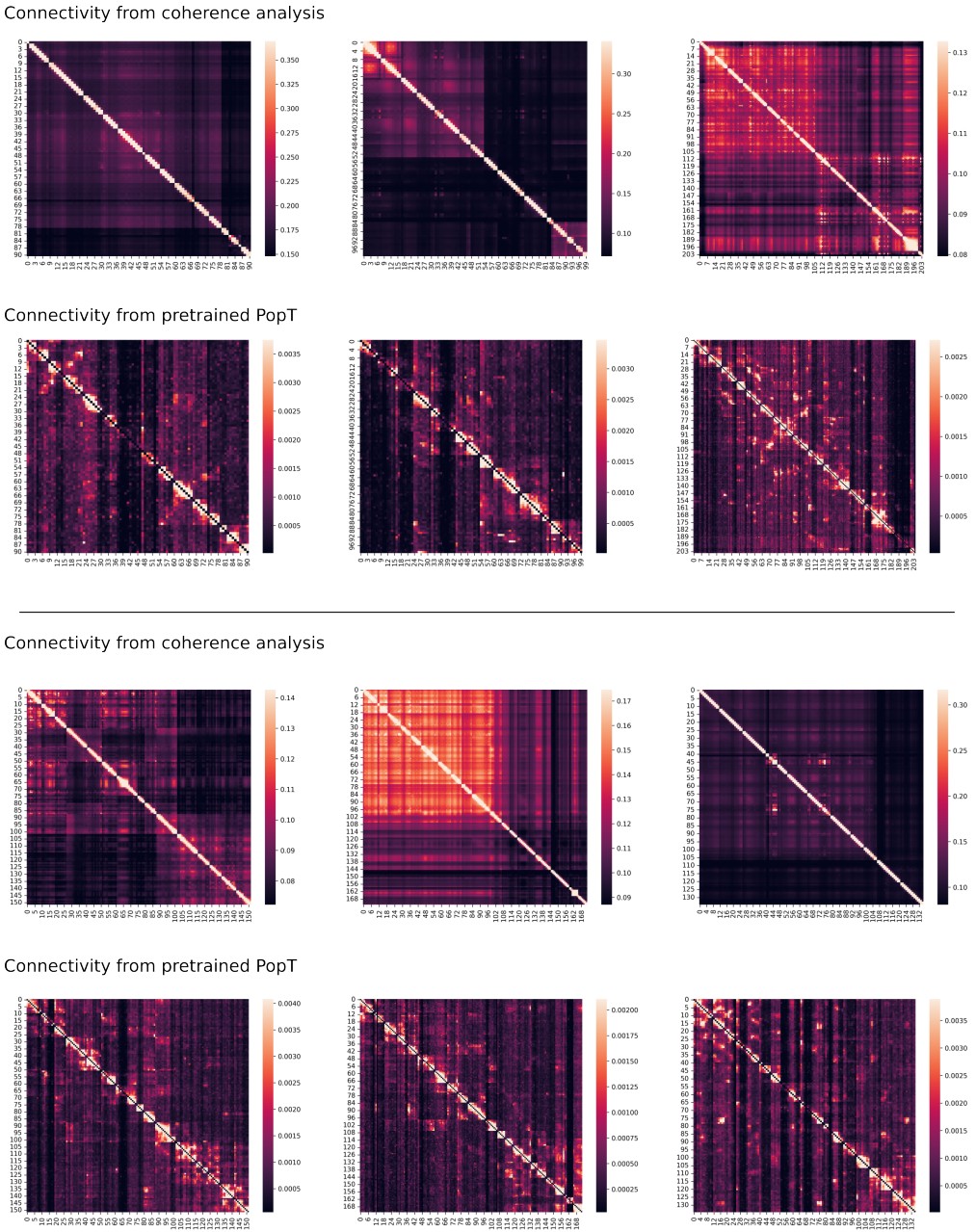

Figure 16: **Electrode connectivity for test subjects.** Continued from Appendix I; this figures shows the rest of the test subjects. Order is given as in Figure 15.

| Subject | Correlation |
|---|---|
| Subject 1 | 0.42 |
| Subject 2 | 0.66 |
| Subject 3 | 0.54 |
| Subject 4 | 0.55 |
| Subject 6 | 0.44 |
| Subject 7 | 0.44 |
| Subject 10 | 0.50 |

Table 7: Pearson's $r$ correlation coefficients between connectivity matrices for test subjects shown in Table 7 and Figure 16.

## J   ADDITIONAL ABLATIONS

|  | Pitch | Volume | Sent. Onset | Speech/Non-speech |
|---|---|---|---|---|
| PopT | **0.74 ± 0.03** | **0.87 ± 0.03** | **0.90 ± 0.01** | **0.93 ± 0.02** |
| PopT w/ gaussian blur | 0.73 ± 0.03 | 0.86 ± 0.03 | 0.88 ± 0.02 | 0.91 ± 0.03 |
| PopT w/o channel randomize | 0.71 ± 0.02 | 0.85 ± 0.03 | 0.89 ± 0.01 | 0.91 ± 0.03 |

Table 8: **PopT additional ablation study.** We pretrain additional variations of PopT to see their effect on downstream decoding. In 'PopT w/ gaussian blur' we fuzz the input coordinate values with a Gaussian, $\mathcal{N}(\mu = 0, \sigma = 5)$, before position encoding. We hypothesized that augmenting the coordinates during training would help the model generalize better, but no improvements were shown. 'PopT w/o channel randomize' replaces a channel with a channel's own activity at another time as part of the channel-wise pretraining task. We hypothesized that this would help the model identify the channel's specific variability across time, but no improvements were shown. Shown are ROC-AUC mean and standard error across subjects evaluated at 90 electrodes.

## K   FROZEN ENSEMBLE SCALING

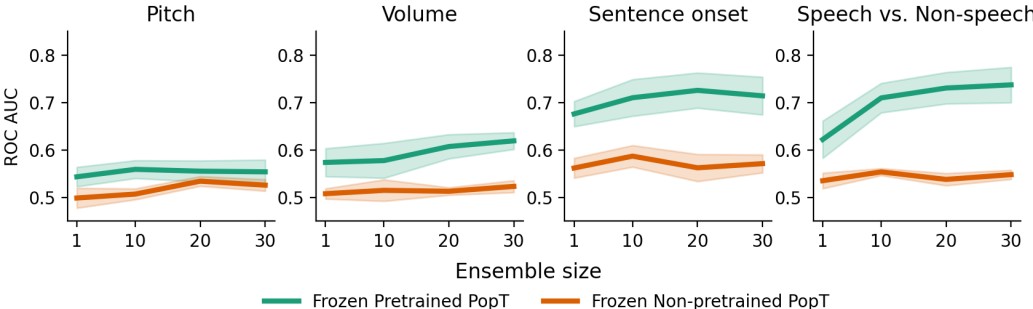

Figure 17: **Pretraining is critical to frozen PopT performance that scales with the number of channels.** Transformer weights are frozen; only the linear classification head is updated during fine-tuning. As in Figure 3, we see that pretraining results in better downstream decoding and scales with the number of added channels. Bands show standard error across subjects.

