# OpenReview forum: "Population Transformer: Learning Population-level Representations of Neural Activity"
_ICLR.cc/2025/Conference — ICLR 2025 Oral_

### Official Review · Reviewer_P8gU · 2024-10-30

**Soundness:** 3
**Presentation:** 2
**Contribution:** 3
**Rating:** 8
**Confidence:** 5

**Summary:**

In this paper, the authors propose a self-supervised learning framework for neural time series data. The model is trained using ensemble and channel-wise discrimination strategies to address the problem of sparse and variable electrode distribution across subjects and datasets. The authors presented the validity of the proposed method on two datasets with diverse analyses.

**Strengths:**

In this paper, the author tried to solve the problem of sparse and variable electrode distribution across subjects and neural time series data datasets by modeling with a single-channel embedding method.

The effectiveness of the proposed method was supported through various experiments and analyses.

It achieved high performance with fewer computational resources compared to existing pre-trained models.

**Weaknesses:**

It is unclear how exactly the position of each channel is defined in spatial position encoding. The reference provides coarse positions only, which is important information in spatial relation learning.
Page 4, line 166: The notations used in position embedding are not defined.

Supporting materials must be presented for the Gaussian fuzzing in channel embedding to ensure its effectiveness in avoiding overfitting.

The experiments are somehow limited. More rigorous and thorough experiments should be conducted over diverse datasets. Other more challenging tasks, e.g., TUEV in the TUH EEG corpus, should be conducted and compared with the comparative methods.

The results in the ablation study (Table 3) suggest that position encoding played a significant role in performance improvements, while other proposed loss functions had marginal effects. In this regard, the related contexts should be provided in detail.

Regarding the connectivity construction in Section 6, it is not straightforward to understand how the connectivity is measured. More detailed explanations and rationales should be provided regarding why the proposed method can estimate channel connectivity.

**Questions:**

Page 3, line 161: According to the paper's description, the input to the temporal encoding is a single value of a channel at a time t. Is this correct?

---

> ### Author Response · Authors · 2024-11-23
>
> Thank you for taking the time to review our work\! We appreciate the comments, which have helped us improve the paper. We address your questions below:
>
> 1. > how exactly the position of each channel is defined in spatial position encoding…The notations used in position embedding are not defined.
>    * Thanks for the question\! We have clarified the text in response to your feedback. Each channel has a 3D coordinate, as given by the datasets from BrainBERT and TUEG respectively. For the iEEG dataset, these coordinates are locations in the brain volume. For the EEG dataset, they are locations on the scalp. We can represent each of these three coordinates as vectors, using the BERT position encoding scheme. We concatenate the three vectors together to get a representation of the spatial position. We have added clarification on the notation to the text.
> 2. > The experiments are somehow limited…more tasks, e.g., TUEV in the TUH EEG corpus, should be conducted.
>    * Since our goal is to study multi-channel aggregation, we evaluate using intracranial and EEG data that contain multi-channel data. The EEG dataset mentioned (TUEV) only contains single channel data. It’s worth noting that most other approaches in this space don’t bother to evaluate using multiple datasets, let alone multiple modalities \[1,2,3,4\]. We evaluate using four audio-linguistic tasks from an intracranial dataset and one abnormal EEG detection task, which together cover a wide range of signal morphologies.
>
> 3. > …not straightforward to understand how the connectivity is measured…More detailed explanations and rationales should be provided
>    * Thanks for your feedback! We’ve revised the description of connectivity (Appendix G), added a diagram (Figure 12), and added a pseudocode algorithm (Algorithm 1). In short, the intuition is this: during pretraining, the model learns a channel-wise task that requires it to predict channel information by relying on the context of surrounding channels. Then, after pretraining has finished, we can use the model to query which pairs of channels are important to each other. We do this by selecting a channel, omitting it from the model input, and then measuring which of the model predictions on the surrounding channels are most affected.
>    * To provide further context/comparison for these connectivity results, we have added the electrode connectivity matrices, which can be seen in Supplementary Figures 13 and 15\. These are given both for our method and a traditional method (coherence/cross-correlation). We have also computed the correlation between these matrices, and these can now be seen in Supplementary table 7\. The average correlation is 0.51.
> 4. > Page 3, line 161: According to the paper's description, the input to the temporal encoding is a single value of a channel at a time t. Is this correct?
>    * $x^t\_i$ does not represent a single value in time, but rather an interval of time that begins at $t$. It is actually a vector in $\\mathbb{R}^{T}$ where $T$ is the number of time samples in the interval. We now clarify this in the text.
>
> ## References
>
> \[1\] Zhang, Daoze, et al. "Brant: Foundation model for intracranial neural signal." *Advances in Neural Information Processing Systems* 36 (2024).
>
> \[2\] Jiang, Weibang, Liming Zhao, and Bao-liang Lu. "Large Brain Model for Learning Generic Representations with Tremendous EEG Data in BCI." *The Twelfth International Conference on Learning Representations*.
>
> \[3\] Ye, Joel, and Chethan Pandarinath. "Representation learning for neural population activity with Neural Data Transformers." *Neurons, Behavior, Data analysis, and Theory* (2021).
>
> \[4\] Wang, Christopher, et al. "BrainBERT: Self-supervised representation learning for intracranial recordings." The Eleventh International Conference on Learning Representations.

---

> > ### Comment · Reviewer_P8gU · 2024-11-27
> >
> > Thank the authors for their hard work and response to my concerns.
> > As the answers mostly resolved my concerns, I raised my score accordingly.

---

> > > ### Author Response · Authors · 2024-12-01
> > >
> > > Thank you for your questions and suggestions which were very helpful in improving our paper.

---

### Official Review · Reviewer_vWCS · 2024-10-31

**Soundness:** 3
**Presentation:** 2
**Contribution:** 4
**Rating:** 8
**Confidence:** 3

**Summary:**

The Population Transformer (PopT) is a self-supervised framework designed to learn brain-wide neural activity representations. The approach aims to address common challenges faced by invasive iEEG and non-invasive EEG neural recordings, namely 1) sparsity of recording channels within the brain and 2) variability in channel positioning and reliability. The authors propose a smart data embedding and combining two loss functions that allow pre-training of the network using data from multiple datasets and individuals. Pre-training is shown to be fundamental to lowering the amount of data necessary for fine-tuning neuroscientific tasks, such as decoding experiments. Finally, the authors show experiments and analyses to support the claim that this framework is interpretable and generalizable.

**Strengths:**

This paper has many merits. The idea and approach, from a computational neuroscience perspective, are novel and very interesting. The proposed framework could benefit the neuroscience community at large: combining data from multiple datasets, and learning brain-wide representations that are not idiosyncratic to a specific individual are key to advancing neuroscience research. Many points made throughout the paper and several experiments are convincing and valuable, and I don’t think much work is required in terms of experiments/compute.

**Weaknesses:**

On the flip side, the paper is, at some points, lacking in precision and clarity, at some points a bit colloquial, and at other points, unnecessary jargon. I think that the paper would be stronger, and more accessible, if the authors put some effort into clarifying and streamlining it. Moreover, the claims of interpretability are currently unclear and potentially overstated.

**Questions:**

**Clarity**:  the paper is not really neat. Moreover, there is some sloppiness in the mathematical notation and technical explanation of the method. Finally, the figures are uneven and quite messy. Follows a list of detailed complaints:
- general lack of dimensionality: in many points, it is not clear what the dimensions are of the variables. for example, lines 160-161, what is the dimension of $X_B^t$? Is the “+” used for concatenation or for an actual sum? What is the output dimensionality of the temporal embedding model $B$? Some of these are in Appendix A, but it would be useful to streamline and clarify.
- $[CLS]$ token: it is stated nowhere that this is the token used for downstream decoding/classification tasks, and what space it belongs to. Is it only binary? Please clarify in the text or figure legend.
- Self-supervised Loss (line 173-197): the losses are described in words, but it would be useful and much clearer to write them in formulas. Right now, this paragraph is a bit messy and somewhat colloquial.
- Fine-tuning (line 199-201): colloquial and relies on details only available in the appendix. It would also be useful to formally define what is meant by a decoding task.
- Figures: Fig 1 is unclear and messy. Why does it go from bottom to top, rather than the opposite? Currently, the “Population Transformer inputs” title of Fig 1a is just above the output, making it very confusing. Moreover, the figure has at least four different font sizes. The “+” used in temporal embedding + 3D position is confusing; does it refer to concatenation or addition. The title of the colorbar near the STTF is nearly illegible. Panels b and c could provide clearer information on the loss. Fig 3-4: font sizes widely different across figures; Fig 3 text is hard to read on print. Same for Figs 6-7-8. Fig 8 left title is partially covered by the plot.

**Interpretability**: the claim of interpretability requires some work to be convincing/useful. For example, the connectivity analyses are far from clear (also after reading the Appendix!). It is not immediately clear how one infers (pairwise) connectivity by masking out the activity of one channel at the time and measuring the degradation in channel-wise loss. Can you add a few details on that? And do we learn anything new from this analysis as compared to previous coherence or cross-correlations analyses? Can we have a few more qualitative and/or quantitative comparisons? Same questions go for the functional brain regions from attention weights. Moreover, what are the possible caveats of inferring connectivity or functional regions from PopT?

**Minor comments**:
- line 153: *adapt* -> adopt?
- lines 132, 158, …: BrainBERT was used but never cited.
- line 163: *sinusoidal position encoding*: is there a simple formula or a couple of words to explain this method without necessarily having to read the cited paper?
- line 171: and $\tilde y_i$ *and(?)* respectively, …
- line 188: …is *again(?)* the binary cross-entropy…
- line 374, 478, 483: Figure X -> (Figure X)

**Additional questions**:
*[These questions popped up while first reading the paper; some got clearer on a second read, some not. I don’t think it’s necessary for the authors to answer all these questions, but hopefully, they’ll be useful for improving discussion and intro, or future experiments, or mybe just food for thoughts.]*

The paper claims to learn representations of neural activity; do the authors mean the final output of the PopT or only the CLS output? Can one also learn a neural representation without using a CLS token? Can this representation be thought of and used as a dimensionality reduction method? Who are the intended users of this method? Can this method be used for non-human invasive research, such as calcium imaging, neuropixels, etc? Is it possible that the discriminative nature of the pre-training objective leads to potentially misleading representations in case of noisy/faulty channels?

---

> ### Author Response · Authors · 2024-11-23
> **Author response part 1**
>
> Thank you for taking the time to make a close reading of our work\! We appreciate the helpful suggestions. We’ve made revisions accordingly:
>
> ## Clarity
>
> * > in many points … it is not clear what the dimensions are of the variables…what is the dimension of $X^t\_B$? Is the “+” used for concatenation or for an actual sum? What is the output dimensionality of the temporal embedding model B?
>   * The temporal embedding model $B$ outputs vectors of dimension $d$. The input to the PopT is a collection of such vectors that have been summed (not concatenated) with their position information. Then, the input $X^t\_B$ can be written as an $S \\times d$ matrix where $S$ is the number of channels. We have revised the text to clarify this and the dimension of other variables.
> * > \[CLS\] token: it is stated nowhere that this is the token used for downstream decoding/classification tasks, and what space it belongs to. Is it only binary? Please clarify in the text or figure legend.
>   * Thanks for pointing this out. We clarify this now in section 3 of the text under “Fine-tuning”.
> * > Self-supervised Loss (line 173-197): the losses are described in words, but it would be useful and much clearer to write them in formulas. Right now, this paragraph is a bit messy and somewhat colloquial.
>   * This is a helpful suggestion\! We now give a complete formal description of the losses in Appendix A.
> * > Fine-tuning (line 199-201): colloquial and relies on details only available in the appendix. It would also be useful to formally define what is meant by a decoding task.
>   * We have re-written this section\!
> * > Figures: Fig 1 is unclear and messy. Why does it go from bottom to top, rather than the opposite?
>   * Thanks for the comments. We’re working on cleaning up Figure 1\. We follow the convention of showing information flow from bottom to top (cf. the original Transformers \[1\] and BERT \[2\] papers).
> * > Currently, the “Population Transformer inputs” title of Fig 1a is just above the output, making it very confusing. Moreover, the figure has at least four different font sizes.
>   * We updated figure 1 for clarity and to standardize font sizes.
> * > The “+” used in temporal embedding \+ 3D position is confusing; does it refer to concatenation or addition?
>   * Addition. We now clarify this in the figure caption.
> * > The title of the colorbar near the STFT is nearly illegible.
>   * Removed colorbar, since we are mainly interested in presenting a schematic
> * > Fig 3-4: font sizes widely different across figures; Fig 3 text is hard to read on print. Same for Figs 6-7-8.
>   * We’ve increased figure size and font sizes\!
>   * Figure 8: We abbreviate the region names, so we can display them at a larger size and add a very visible legend
> * > Fig 8 left title is partially covered by the plot.
>   * Fixed\! Thanks for the catch\!

---

> ### Author Response · Authors · 2024-11-23
> **Author response part 2**
>
> ## Interpretability
> - > It is not immediately clear how one infers (pairwise) connectivity by masking out the activity of one channel at the time and measuring the degradation in channel-wise loss. Can you add a few details on that?
>   - Thanks for your feedback\! Given your feedback, we’ve revised the description of connectivity (Appendix G), added a diagram (Figure 12), and added a pseudocode algorithm (Algorithm 1). In short, the intuition is this: during pretraining, the model learns a channel-wise task that requires it to predict channel information by relying on the context of surrounding channels. Then, after pretraining has finished, we can use the model to query which pairs of channels are important to each other. We do this by selecting a channel, omitting it from the model input, and then measuring which of the model predictions on the surrounding channels are most affected.
> - > And do we learn anything new from this analysis as compared to previous coherence or cross-correlations analyses? Can we have a few more qualitative and/or quantitative comparisons?
>   - To provide further context/comparison for these connectivity results, we have added the electrode connectivity matrices, which can be seen in Supplementary Figures 13 and 15\. These are given both for our method and a traditional method (coherence/cross-correlation). We have also computed the correlation between these matrices, and these can now be seen in Supplementary table 7\. The average correlation is 0.51. It seems that the methods arrive at overlapping pictures of connectivity, especially along the strongest points. But in general the PopT seems to discover sparser connectivity maps.
> - > Same questions go for the functional brain regions from attention weights. Moreover, what are the possible caveats of inferring connectivity or functional regions from PopT?
>   - The authors of the original dataset released analyses where they identify word responsive electrodes (see fig 2h-i in \[3\]) using t-tests between pre- and post- word-onset activities. We find a correspondence between regions with a high fraction of word responsive electrodes and the regions with high attention weight, as found by our analyses. We find a Pearson correlation coefficient of 0.4 against the fraction of significant electrodes found in the prior work. It would be good future work to investigate the additional electrodes found to be highly attended to in a trained PopT to evaluate what features may be additionally leveraged for improved decoding.

---

> ### Author Response · Authors · 2024-11-23
> **Author response part 3**
>
> ## Additional questions
> - > The paper claims to learn representations of neural activity; do the authors mean the final output of the PopT or only the CLS output? Can one also learn a neural representation without using a CLS token?
>   - Both the token outputs and the CLS output are meant to be included when we discuss the resulting representations. The CLS token is used for our decoding experiments and the token outputs are used in the connectivity analysis.
> - > Can this representation be thought of and used as a dimensionality reduction method?
>   - Yes\! Good question\! The original BrainBERT paper explores the intrinsic dimension of the learned single channel representations, and this is what we would like to do with our multi-channel representations in the future.
> -  > Who are the intended users of this method?
>    - Neuroscientists and brain machine interface (BMI) researchers: for use in decoding and interpreting brain data.
> - > Can this method be used for non-human invasive research, such as calcium imaging, neuropixels, etc?
>   - For the modalities proposed, because all the activity is recorded from neurons that are physically close together, it’s unclear if the 3D position will be meaningful information. This matters because we find that 3D position is critical information for iEEG and EEG decoding, so it is an open question as to whether this approach would translate effectively.
> - > Is it possible that the discriminative nature of the pre-training objective leads to potentially misleading representations in case of noisy/faulty channels?
>   - There’s merit to this concern. Whereas a generative model would (likely) not learn to produce outputs that look like noise, for a discriminative approach, there’s a chance that out-of-distribution faulty channels could land in the same latent space region as non-faulty channels. However, this becomes less likely as the amount of pretraining data grows, and more kinds of noise channels become attested to in the data. Generally, noisy channels also provide less information when performing our SSL tasks, so we expect that our model to rely less on temporal embeddings that look like noise.
>
> ### Minor comments
> - >line 163: *sinusoidal position encoding*: is there a simple formula or a couple of words to explain this method without necessarily having to read the cited paper?
>    - Yes\! We include a brief description
>  - Thanks for the other catches!
>
> ## References
>
> \[1\] Vaswani, A. "Attention is all you need." *Advances in Neural Information Processing Systems* (2017).
>
> \[2\] Devlin, Jacob. "Bert: Pre-training of deep bidirectional transformers for language understanding." *arXiv preprint arXiv:1810.04805* (2018).
>
> \[3\] Wang, Christopher, et al. "Brain Treebank: Large-scale intracranial recordings from naturalistic language stimuli." NeurIPS (2024).

---

> > ### Comment · Reviewer_vWCS · 2024-11-26
> >
> > The authors took the comments seriously and significantly improved the presentation of the paper. I stand by my initial assessment (8).
> > Thank you for your response and work!

---

> > > ### Author Response · Authors · 2024-11-27
> > >
> > > Thank you for your detailed comments and suggestions. They were very valuable in improving our paper!

---

### Official Review · Reviewer_b3HQ · 2024-11-02

**Soundness:** 3
**Presentation:** 1
**Contribution:** 3
**Rating:** 5
**Confidence:** 2

**Summary:**

This paper introduces a self-supervised framework, the Population Transformer (PopT), designed to learn population-level representations for large neural recording datasets.  PopT addresses challenges related to sparse and variable electrode distribution across subjects and datasets by using pretrained embeddings. By using a modular approach the model is more flexible and lightweight. The authors claim that the approach is computationally efficient, interpretable, and performs competitively against end-to-end methods.

**Strengths:**

This paper demonstrates strong rigor by evaluating the proposed method across two types of neural time-series data (iEEG and EEG), enhancing the generalizability and robustness of its findings. The authors plan to share both the data and code upon acceptance, promoting transparency and reproducibility within the community. Additionally, they test their method using hold-out subject tests to validate the model’s performance on unseen data, which is essential for assessing real-world applicability. The method’s effectiveness is benchmarked against a diverse set of models, providing a comprehensive view of its performance relative to other approaches. Furthermore, ablation studies are conducted to examine the contribution of different components in the proposed framework, offering valuable insights into how each part enhances overall model performance.

**Weaknesses:**

The study offers a comprehensive evaluation of PopT, and assess it using various experiments. However, the paper would benefit from clearer organization and writing, particularly in emphasizing the significance of each experiment.

- **Introduction and Field Context**: Despite the extensive experimentation, the paper lacks a clear introduction to the field and a focused statement of its goals. Important background information and foundational definitions are often buried in the appendix rather than integrated into the main text.
    - For instance, a key innovation of the study is the use of channel-level and ensemble objectives. Providing a more detailed literature review on current approaches to these objectives would help the reader understand the limitations of existing methods and the advantages of the proposed improvements. This contextualization would make the study’s contributions clearer and more impactful.
    - The connection between experiments is also unclear, with critical information about the pretrained PopT model only appearing in the appendix. Providing a clear description of the data used for pretraining within the main text would help readers understand the study’s foundation and goals more effectively.
    - Similarly, Table 3 introduces PopT with an L1 reconstruction loss as an additional experiment. However, this experiment is not discussed in the text, and it appears tangential to the core contributions of the study. Omitting such details could allow space to expand on more relevant analyses.
- **Benchmarking Choices**: While Chronos/TS2Vec and BrainBERT/TOTEM are used as benchmarks for PopT, it remains unclear why these specific models were selected. The authors could strengthen the study by discussing the criteria for choosing these benchmarks.
- **Minor Presentation Issue**: In Figure 8, the plot slightly overlaps with the title on the left image.

**Questions:**

1. Table 3 refers to “PoPT w/o group-wise loss”, is the group-wise loss the same of ensemble-wise loss. To improve clarity, could the authors consider using consistent terminology throughout?
2. It would be insightful to see how one of the baseline models, specifically BrainBERT from Table 1, performs on the hold-out dataset, similar to the performance reported for PoPT in Figure 6. This comparison would provide additional context for the robustness of PoPT relative to other methods. Could the authors report the performance of BrainBERT on the hold-out dataset in a format similar to Figure 6?

---

> ### Author Response · Authors · 2024-11-23
>
> Thank you for your review of our work\! We appreciate the feedback and address the comments below:
>
> 1. > Despite the extensive experimentation, the paper lacks a clear introduction to the field and a focused statement of its goals. Important background information and foundational definitions are often buried in the appendix rather than integrated into the main text.
>    * Thanks for the feedback\! We’ve revised our introduction to start with a clear statement of our motivation and goals, with care to explain the relevant prerequisite definitions.
> 2. > For instance, a key innovation of the study is the use of channel-level and ensemble objectives. Providing a more detailed literature review on current approaches to these objectives would help the reader understand the limitations of existing methods and the advantages of the proposed improvements.
>    * We will emphasize the field’s exploration into channel-level aggregation in the related works and background. To our knowledge, most deep learning approaches take an end-to-end training approach as outlined in our Related Works, and only a few have explored channel-level aggregation. Those that do focus on supervised training, rather than tackling the problem of leveraging unannotated data from variable subject layouts.
> 3. > The connection between experiments is also unclear…a clear description of the data used for pretraining.
>    * Given your feedback, we’ve revised and done our best to fit more information about the data and PopT into the main text.
> 4. > L1 reconstruction loss…is not discussed in the text…appears tangential to the core contributions…\[suggest\] omitting
>    * Thanks\! We now explicitly discuss this in the text. The experiment shows the necessity of a discriminative loss, rather than a reconstructive loss, as is used in other self supervised approaches, e.g. BrainBERT. Since we consider this an important point, and given that it is only one line, we would like to keep this result in the main text\! But given your feedback, we will remove the closely-related experiment (L1 \+ discriminative loss) since it is mostly redundant.
> 5. > While Chronos/TS2Vec and BrainBERT/TOTEM are used as benchmarks for PopT, it remains unclear why these specific models were selected. The authors could strengthen the study by discussing the criteria for choosing these benchmarks.
>    * We have revised to add a brief description/reason for including each encoding model. In short, they cover a wide range of encoding motifs: convolutional (TS2Vec), tokenizing (TOTEM), transformer based (Chronos), and iEEG specific (BrainBERT).
> 6. > In Figure 8, the plot slightly overlaps with the title on the left image
>    * Thanks for the catch\! Fixed.
> 7. > Table 3 refers to “PoPT w/o group-wise loss”, is the group-wise loss the same of ensemble-wise loss. To improve clarity, could the authors consider using consistent terminology throughout?
>    * Thanks for the catch\! Yes. They are meant to refer to the same thing. We have updated the text to be consistent.
> 8. > It would be insightful to see how one of the baseline models, specifically BrainBERT from Table 1, performs on the hold-out dataset. Could the authors report the performance of BrainBERT on the hold-out dataset in a format similar to Figure 6?
>    * This is a very helpful suggestion\! The BrainBERT performances on the test dataset have been added to Figure 6\. We also run the same hold-one-out analysis on a PopT trained with TOTEM, which we show in Appendix E.

---

### Official Review · Reviewer_LNxL · 2024-11-03

**Soundness:** 3
**Presentation:** 3
**Contribution:** 3
**Rating:** 8
**Confidence:** 3

**Summary:**

The authors introduce Population Transformer (PopT), a transformer-based and contrastive learning approach for decoding neural time series data. The PopT aggregates and decodes channel-wise temporal encodings of neural time series data from, e.g., a pretrained BrainBert but is agnostic to the encoder. The authors introduce two contrastive loss terms for training the PopT. One contrastive loss term requires the PopT to learn temporal representations for each channel, and the other contrastive loss term requires the PopT to learn spatial representations across channels. The PopT receives electrode placement as input, allowing it to decode neural activity across varying electrode placements (e.g., from several subjects). In an application with EEG and iEEG, the paper presents that the PopT converges with fewer samples than other shown methods, achieves superior decoding performance compared to other shown methods, and provides evidence for the PopT's ability to generalize across subjects. The authors also suggest measuring and interpreting functional connectivity based on the ablation of individual channel input to their PopT and performance degradation for all other channels. They also suggest interpreting the weights of the model to map the task-associated function to specific electrodes.

**Strengths:**

The claimed contributions of the PopT would present an exciting leap forward in decoding neural time series data. This is because unreliable and slow decoding of neural activity limits applications and scientific insights from models. A method that requires comparably few data, no labeled data, and learns representations that generalize across subjects regardless of electrode layout makes decoding more fast and reliable for new subjects and unlocks new applications. The ability of the PopT to generalize across subjects is a major strength of the approach. Another strength is that the PopT relies on pre-trained encoding models that it is agnostic to. The formulations of the contrastive learning objectives seem original. The suggested ways to interpret the model to understand the data are interesting.

**Weaknesses:**

The paper presents multiple weaknesses that could be addressed to improve its clarity and reliability.

First, the abstract describes PopT's benefits, such as "computationally lightweight" and "more interpretable," which lacks precision without clear comparison groups or quantitative metrics. For example, calling PopT computationally lightweight in comparison to end-to-end trained models is misleading since it relies on pre-trained, e.g., BrainBert weights, which already bear significant computational demands. As another example, the interpretability analysis does not include any comparison method.

In addition, the generalizability claims lack thorough validation: the hold-one-out subject analysis does not sufficiently establish robustness. This experiment could benefit from a broader validation, e.g., cross-validation approaches.

Additionally, when comparing the performances to other models the analysis does not account for the differences in free parameters and nonlinearities across models.

Last, the codebase requires significant improvements, as it is messy, poorly documented, and lacks proper instructions or tutorials. For example, the readme is incorrectly titled BrainBert 2.0. This contrasts the authors claim of it being "plug-and-play."

I am not up-to-date and fully familiar with related work. This is why my expertise might be insufficient to fully weigh the significance of the central contribution against the current limitations of the paper in the clarity and reliability of the results.

I am generally excited about the direction of the work and happy to give a higher evaluation once the presentation of the current version of the paper is significantly improved and the detailed questions below are addressed.

**Questions:**

# **Major Questions**

1. You show that the decoding performance increases with more subjects and decreases with holding out a subject. However, these experiments are limited because they consider a validation on specific channels of one specific subject, or holding out one specific subject. Is this also valid across arbitrary channels and arbitrary subjects?
2. Figure 4: Do the baseline aggregation approaches have fewer free parameters and nonlinearities than the PopT? Could you report the number of free parameters and nonlinearities of all networks and factor this into the discussion of your results?
3. Results 366-370: The model's ability to generalize is a central claim. However, the hold-one-out analysis seems somewhat limited. What if the held-out subject was a lucky draw? Can this comparison be done in a crossvalidation? Can the performance decrease be computed for hold-k-out and presented as a function of k? Separately, is there a different model class that can generalize to provide a comparison group in this test?
4. 371-377 and Figure 6 and 7: I would have expected that "All subjects" and "0 subjects" correspond to "All" and "Non-pre-trained PopT" but the numbers are different. Can I rely on this data? Why is there a mismatch?
5. Figure 7: What if "across channel ensembles 5-30 on a held out test subject" is a lucky draw? Could this be more robustly supported with an extended analysis?
6. 414-416: To say this, I would expect an analysis of representations resulting from different loss functions. Can one show that the contrastive learning objective results in the clearest association/separation of inputs?
7. 468-470: One obtains N-squared performances, where N is the number of channels. Can one present the raw functional connectivity matrices inferred this way in addition to the aggregate? This could help to gauge how effective the approach is.
8. 468-470: Can one compare the results to the functional connectivity inferred with traditional cross-correlation methods on the measurements? This would help to illustrate the true utility of the PopT-based analysis.
9. 475-485: Same here. Can one compare to neuroscience ways to characterize the activated brain area, e.g., z-score based?
10. Results 296-297: What are differences in, e.g., architecture or training that makes LaBraM outperform the PopT? Could one provide possible explanations for why LaBraM outperforms the PopT on EEG decoding (whereas end-to-end approaches specifically designed for iEEG do not)?
11. Figure 4: there is a much more significant improvement for the sentence onset and speech vs. non-speech tasks over the improvement in the pitch and volume tasks. Could one provide an explanation for this in the results section?
12. In contrast to one of the central claims in the abstract, 23-24, the codebase is messy, and lacks documentation, clear examples, instructions, and tutorials. The readme title is BrainBert 2.0. Could the codebase please be significantly cleaned and documented?

# **Clarifications and Minor Questions**

13. Abstract 16-18: The sentence "lowers the amount of data for downstream decoding experiments" is compatible with the presented results (Fig. 4 and 5). However, the verb 'lowers' requires a comparison group, and 'downstream decoding experiments' is vague. The meaning of 'downstream' is also ambiguous. Could you please rephrase this?
14. Abstract 18-20: Calling the PopT computationally lightweight in comparison to end-to-end approaches is misleading because the PopT only works with pretrained weights from BrainBert. Could you please rephrase this?
15. Abstract 18-20: Calling the PopT "more interpretable" when not comparing the interpretations to existing traditional neuroscientific methods for interpretability is misleading. Could you please rephrase this?
16. Abstract 18-20: What does 'retaining competitive performance' mean? In all iEEG data, it outperforms the end-to-end method used, in all EEG data it is outperformed by LaBraM. Could you please provide this information without ambiguity?
17. Abstract 21-23: What are "massive" amounts of data according to the authors?
18. Discussion 524-525: "\[ … \] could provide an even higher resolution latent space \[ … \]". Higher than what? Could you please rephrase this?
19. Conclusion 538-539: I wouldn't know how to use it "for plug-and-play pretraining". The codebase is messy and lacks clear examples, instructions, or tutorials. Is this a work in progress or the cleaned-up codebase?
20. 206-208: The explanation is not reader-friendly and not comprehensible for someone who does not know the types of neurorecordings. Could you please explain to the reader why intracranial and scalp electroencepholography lead to two different types of time-series data, and why and which resolution extremes these represent?
21. 3 Population Transformer Approach 178-179: these are rather "contrastive" components and not "discriminative" components. Could one please adopt "contrastive" instead of discriminative throughout the text?
22. Figure 2, 6, 7: Could one use violin (or box) plots instead of bar plots to convey the variability and scatter the individual data points on top to convey the sample size?
23. 375: "potentially due to adaption to the temporal embeddings used" meaning unclear. Could you please rephrase this?
24. 363-365: These sentences seem contradictory, which might be easily resolved with rephrasing them. Could you please rephrase this? Part of the Figure 5 caption could go here into the main text.
25. Results Figure 4: What is 'highly' sample efficient?

# **Minor Formatting**

26. 214-215: Could you please remove the parenthesis inside the parenthesis?
27. Table 1: Could you please emphasize your result better by bolding the label Pretrained PopT or star it for "best overall"?
28. Results 264-269: hard to read. Could you please remove the parenthesis inside the parenthesis?
29. Figure 3 is referred to before Figure 2 in the text. Could you please renumber the figures to maintain clarity?
30. Table 1 and Figure 1: Could you please make latex keep them inside the section where they are referred to?

---

> ### Author Response · Authors · 2024-11-22
> **Author response part 1**
>
> Thank you for taking the time and effort in reviewing our work\! We truly appreciate the feedback and opportunity to further improve the paper.
>
> 1. > calling PopT computationally lightweight in comparison to end-to-end trained models is misleading since it relies on pre-trained, e.g., BrainBert weights, which already bear significant computational demands.
>    * In fact, this is exactly what we mean by “computationally lightweight”\! Our main argument is that the research community has already produced many pre-trained weights for single-channel temporal embeddings, and this means there is an opportunity to cheaply learn multi-channel embedding by aggregating across single-channel representations. Our presented approach is lightweight compared to end-to-end models which require backpropagation through the entire temporal encoding and spatial encoding stack. To make this point precise, we show a comparison table of trainable parameters between PopT and other end-to-end models in Appendix B.
> 2. > "more interpretable," … lacks precision without clear comparison groups or quantitative metrics… the interpretability analysis does not include any comparison method…
>    * By “more interpretable” we just mean to say that end-to-end models lack an attention weight matrix across individual channel units. But for the sake of strict correctness, we will drop the word “more” since we do not make an explicit comparison with end-to-end model interpretability methods.
> 3. > In addition, the generalizability claims lack thorough validation: the hold-one-out subject analysis does not sufficiently establish robustness. This experiment could benefit from a broader validation, e.g., cross-validation approaches.
>    * We apologize for the confusion that resulted from our sparse text description. The hold-one-out results are, in fact, cross-validated. That is, we do not draw a single subject to test on, but instead, we hold out all test subjects in turn from pretraining and then evaluate on the held out subject. The error bars reflect the variation across held-out subjects. We now clarify this in the text.
> 4. > the codebase requires significant improvements, as it is messy, poorly documented, and lacks proper instructions or tutorials.
>    * We have cleaned up the codebase and it can be seen in the updated supplementary materials. Since we are actively developing the code, it does remain a work in progress. The codebase, with documentation and tutorials, will be complete and ready for public release by the time of the conference. As it stands, we have made the code base neater, and outlined the minimal set of commands needed to run PopT.

---

> ### Author Response · Authors · 2024-11-22
> **Author response part 2**
>
> # Responses to Major Questions:
>
> 1. > You show that the decoding performance increases with more subjects and decreases with holding out a subject. However, these experiments are limited because they consider a validation on specific channels of one specific subject, or holding out one specific subject. Is this also valid across arbitrary channels and arbitrary subjects?
>    * > However, these experiments are limited because they consider…one specific subject.
>       *  We respond to this more fully in addressing your points 3,4, and 5 (see below). But in short, the hold-one-out analysis from Figure 6 actually does show cross-validated results, not the results for one specific subject. We have made this clearer in the text now.
>    * > Is this also valid across arbitrary channels …?
>       * PopT takes multiple channels as input, and all subjects are each eventually evaluated on the full 90 channels of input (see the rightmost end of each line in Figure 3). One could still be concerned that we have gotten lucky with our specific *ordering* of electrode subsets.To this end, we have added a plot in Appendix D which shows results for randomly selected electrode subsets.
> 2. > Could you report the number of free parameters and nonlinearities of all networks and factor this into the discussion of your results?
>    * We provide a comparison of the free parameters with our baseline models and SOTA deep learning models in Appendix B Table 4\. The DeepNN baseline and our PopT contain a similar number of trainable parameters. Both the DeepNN and the PopT use non-linear GeLU activation functions.
> 3. > What if the held-out subject was a lucky draw? Can this comparison be done in a crossvalidation?...Can the performance decrease be computed for hold-k-out and presented as a function of k?...I would have expected that "All subjects" and "0 subjects" correspond to "All" and "Non-pre-trained PopT" but the numbers are different…Figure 7: What if "across channel ensembles 5-30 on a held out test subject" is a lucky draw?
>    * We address your points 3, 4, and 5 as a unit. These are valid concerns. And we see that our sparse descriptions have caused understandable confusion. First, to clarify the status quo:
>       * The hold-one-out results in Figure 6 are, in fact, cross-validated. That is, we do not draw a single subject to test on, but instead, we hold out all test subjects in turn from pretraining and then evaluate on the held out subject. The error bars reflect the variation across test subjects. We now clarify this in the text.
>       * The scaling results in the (old) paper’s Figure 7 were not cross-validated. We draw a single subject to test on, and progressively add more subjects to the training. All evaluation is done on this single subject. This is the reason why the numbers for “0 subjects” and “Non-pretrained PopT” do not match between Figures 6 and 7\. We used this design, because it was not feasible to cross-validate when holding out more subjects, since each pretraining run takes about half a week, and the number of pre-training runs that are required for an exhaustive hold-k-out analysis grows combinatorially as a function of k.
>    * In the case of the scaling results, your concern about a lucky draw is fair.
>       * To this end, we have run a new experiment in which we create truncated datasets to pretrain on. You can see the updated figure in the revised paper’s Figure 7\. Now, since the independent variable is the percentage of pretraining data, rather than the number of subjects, we do not encounter the combinatorial problem described above. The performance is now evaluated on all test subjects, the same as in the hold-one-out results.
> 4. See 3.
> 5. See 3.

---

> ### Author Response · Authors · 2024-11-22
> **Author response part 3**
>
> 6. > To say \[more informative\], I would expect an analysis of representations resulting from different loss functions.
>    * By “informative” we only mean “more beneficial for decoding,” as evidenced by the result of the ablation experiments. We have revised the text to clarify this.
> 7. > Can one present the raw functional connectivity matrices inferred this way in addition to the aggregate?
>    * Done\! The raw channel connectivity matrices can now be seen in Appendix H: Connectivity.
> 8. > Can one compare the results to the functional connectivity inferred with traditional cross-correlation methods on the measurements
>    * The correlation between the connectivity matrices obtained via PopT and the matrices obtained via coherence analysis can now be seen in Appendix H: Connectivity. The average Pearson’s *r* correlation across test subjects is 0.51.
> 9. > Can one compare to neuroscience ways to characterize the activated brain area?
>    * The authors of the original dataset released analyses where they identify word responsive electrodes (see fig 2h-i in \[1\]) using t-tests between pre- and post- word-onset activities. We find a correspondence between regions with a high fraction of word responsive electrodes and the regions with high attention weight, as found by our analyses. We find a Pearson correlation coefficient of 0.4 against the fraction of significant electrodes found in the prior work.
> 10. > What are differences in, e.g., architecture or training that makes LaBraM outperform the PopT? Could one provide possible explanations for why LaBraM outperforms the PopT on EEG decoding (whereas end-to-end approaches specifically designed for iEEG do not)
>     * LaBraM is designed specifically for EEG, so it is not surprising that it performs well in this domain. LaBraM leverages EEG specific information such as power spectral densities as part of the temporal encoding, and was pretrained on a wide variety of EEG data. It is rather surprising that PopT remains competitive with LaBraM, despite not being end-to-end, nor specialized for EEG (we only pretrain on a single EEG dataset and use general off-the-shelf time-series encoders).
>     * Although Brant was specifically designed for iEEG, it was trained on lower sampling rate data and only leverages MLP aggregation of electrode information. These design choices may have hindered its ability to perform on our more challenging decoding tasks.
> 11. > there is a much more significant improvement for the sentence onset and speech vs. non-speech tasks over the improvement in the pitch and volume tasks. Could one provide an explanation for this in the results section?
>     * Regions of cortex are functionally specialized, so the sampling of electrodes will determine the decodability for each task. The electrodes in the intracranial dataset are sampled heavily from speech processing areas, namely the superior temporal gyrus. This performance trend can also be seen in the original BrainBERT paper’s findings for the frozen BrainBERT embeddings, for which the Sentence onset and Speech vs. Non-Speech had a mean ROC-AUC of 0.66 and 63, while the Pitch and Volume tasks had 0.51 and 0.60. We revised the text to mention this.
> 12. Codebase is cleaned and updated (see new zip file and author response part 1 for more elaboration).

---

> ### Author Response · Authors · 2024-11-22
> **Author response part 4**
>
> # Clarifications
>
> 13. > Abstract 16-18: The sentence "lowers the amount of data [required] for downstream decoding experiments" is compatible with the presented results (Fig. 4 and 5). However, the verb 'lowers' requires a comparison group, and 'downstream decoding experiments' is vague. The meaning of 'downstream' is also ambiguous. Could you please rephrase this?
>     - Here, the implicit comparison group is "the amount of data that would otherwise be needed" in the models we compare against.
>     - "Downstream" and "downstream tasks" are standard terms when discussing foundation models (cf. \[2\])
> 14. > Calling the PopT computationally lightweight in comparison to end-to-end approaches is misleading because the PopT only works with pretrained weights from BrainBert.
>     - PopT can be trained on top of any pretrained temporal embedding, not just BrainBERT. In our experiments, we pretrained with TOTEM, TS2Vec, and Chronos.
> 15. > Abstract 18-20: Calling the PopT "more interpretable" when not comparing the interpretations to existing traditional neuroscientific methods for interpretability is misleading. Could you please rephrase this?
>     - This is fair. We mean "interpretable" as demonstrated by our study of the attention weights and connectivity. For strict correctness, we will remove the word "more" since we did not make an explicit comparison with other methods.
> 16. > Abstract 18-20: What does 'retaining competitive performance' mean? In all iEEG data, it outperforms the end-to-end method used, in all EEG data it is outperformed by LaBraM. Could you please provide this information without ambiguity?
>     - Here we simply mean that PopT's performance is always at least in the ballpark of recent end-to-end methods. We have revised our abstract to say "similar or better performance".
> 17. > What are "massive" amounts of data?
>     - The dataset we use contains 4,551 electrode-hours of recordings (see original BrainBERT paper). For context, it's rare to have this amount of intracranial data for the movie watching setting. The most similar dataset to the dataset we use, Berezutskaya et al. \[3\] , presents participants with vastly less stimuli: a 6.5 minute short movie, compared to the average of 4.3 hours of movie per subject in the data we use.
> 18. > Discussion 524-525: "\[ … \] could provide an even higher resolution latent space \[ … \]". Higher than what? Could you please rephrase this?
>     - Sorry; that was confusing as previously written. We've revised that section to clarify.
> 19. > The codebase is messy and lacks clear examples, instructions, or tutorials. Is this a work in progress or the cleaned-up codebase?
>     - We respond to this point above and in point 12\. In short, we have cleaned up the codebase and it can be seen in the updated supplementary materials. As we are actively developing the code, it remains a work in progress.The codebase, with documentation and tutorials, will be complete by the time of the conference. As it stands, we have made the code base neater, and outlined the minimal set of commands needed to run PopT.
> 20. > The explanation is not reader-friendly and not comprehensible for someone who does not know the types of neurorecordings.
>     - Thanks for the feedback\! We've revised that section to explain the differences between the types of neurorecordings in more detail.
> 21. > these are rather "contrastive" components and not "discriminative" components.
>     - We use the descriptor "discriminative" to reflect the fact our objective function formula is given in terms of classification, i.e., a single prediction is made for each single input token. In comparison, the objective function formula for a contrastive approach like SimCLR takes positive and negative examples as inputs and explicitly contrasts them using an interaction term.
> 22. > Figure 2, 6, 7: Could one use violin (or box) plots instead of bar plots to convey the variability and scatter the individual data points on top...?
>     - We have created updated Figures 6 and 7\! We keep Figure 2 as a bar plot for visual clarity, since it does not fit on the page as a violin/box plot.
> 23. > "potentially due to adaption to the temporal embeddings used" meaning unclear. Could you please rephrase this?
>     - We have re-written that entire section following the feedback on points 3,4,5.
> 24. > These sentences seem contradictory, which might be easily resolved with rephrasing them. Could you please rephrase this? Part of the Figure 5 caption could go here into the main text.
>     - Yes, that was confusing as it was written. We have revised that section.
> 25. > What is 'highly' sample efficient?
>     - We just mean that the PopT can achieve the full performance of the baseline models after having seen $\\approx 10%$ of the dataset.

---

> ### Author Response · Authors · 2024-11-22
> **Author response part 5**
>
> # Minor Formatting
>
> 26. > 214-215: Could you please remove the parenthesis inside the parenthesis?
>     - Done\!
> 27. > Figure 3 is referred to before Figure 2 in the text. Could you please renumber the figures to maintain clarity?
>     - Done\! We reordered the sentences in the text so that the figures are referred to in order.
> 28. > Table 1: Could you please emphasize your result better by bolding the label Pretrained PopT or star it for "best overall"?
>     - Done\! See updated text.
> 29. > Results 264-269: hard to read. Could you please remove the parenthesis inside the parenthesis?
>     - Done\! See updated text.
> 30. > Table 1 and Figure 1: Could you please make latex keep them inside the section where they are referred to?
>     - Done\! Table 1 and Figure 1 are now in their respective sections where they are first mentioned.
>
> # References
> \[1\] Wang, Christopher, et al. "Brain Treebank: Large-scale intracranial recordings from naturalistic language stimuli." *NeurIPS* (2024).
> \[2\] Bommasani, Rishi, et al. "On the opportunities and risks of foundation models." *arXiv preprint arXiv:2108.07258* (2021).
> \[3\] Berezutskaya, Julia, et al. "Open multimodal iEEG-fMRI dataset from naturalistic stimulation with a short audiovisual film." *Scientific Data* 9.1 (2022): 91\.

---

> ### Comment · Reviewer_LNxL · 2024-11-27
>
> The authors have addressed my concerns and questions rigorously and convincingly. I have increased my evaluation to 8: accept, good paper. Thank you for your hard work.

---

> > ### Author Response · Authors · 2024-11-27
> >
> > We appreciate your thorough feedback and are confident that the discussion and detailed suggestions have enhanced our paper's quality.

---

### Official Review · Reviewer_VeNz · 2024-11-04

**Soundness:** 3
**Presentation:** 3
**Contribution:** 3
**Rating:** 8
**Confidence:** 2

**Summary:**

The authors introduce a self-supervised learning framework called PopT to learn embeddings for EEG and iEEG recordings for the improving downstream decoding analysis. In general, the problem is hard because of the spatially varying channel placements in different experiments. Their method leverages unannotated data to optimize channel-level and ensemble-level objectives, which helps them build generic representations and also allows to capture some dynamical relationships. Their tests show improved decoding on held-out subjects. Based on the interpretation of the weights, they propose a new method for brain region connectivity analysis and for identifying candidate brain regions.

**Strengths:**

- They tackle an important and difficult problem for the field

- Interesting approach backed with strong empirical results.

- I appreciate the work authors put in doing new analyses and interpreting the weights.

-Well written and interesting.

**Weaknesses:**

I have some questions about their connectivity analyses below

**Questions:**

In Fig. 8, how does the connectivity shown compare with cross-correlation or some other metrics that are used in the field?

---

> ### Author Response · Authors · 2024-11-22
>
> Thank you for your response! We appreciate the interest in the problem space and your question regarding connectivity metrics and analysis:
>
>
> * > …how does the connectivity shown compare with cross-correlation…?
>     * For comparison, we include traditional coherence analysis, which is simply cross-correlation, taken across different frequency bands and then averaged. This is shown on the left hand side of Figure 8. We have added more typical channel-connectivity matrices in Supplementary Figures 13 and 15. We show these both for our method and for cross-correlation (coherence).  For each pair of matrices, we measure the correlation between our method and the traditional method. These can be seen in our newly added Table 7. The average Pearson correlation coefficient is 0.51.

---

> > ### Comment · Reviewer_VeNz · 2024-11-27
> >
> > The new appendix and analyses comparing coherence-based measure with author's connectivity measure is much appreciated and highlights well the advantage of their method.  With that the authors have resolved all my concerns.

---

> > > ### Author Response · Authors · 2024-11-27
> > >
> > > Thank you for your feedback. We appreciate the time taken to review our paper!

---

### Official Review · Reviewer_jUF4 · 2024-11-04

**Soundness:** 3
**Presentation:** 3
**Contribution:** 3
**Rating:** 8
**Confidence:** 3

**Summary:**

The manuscript introduces the Population Transformer (PopT). This self-supervised framework tackles two key challenges in neural time-series analysis: sparse electrode distribution and variable electrode placement across subjects and datasets. PopT uses a transformer architecture that combines temporal embeddings with spatial information and trains through self-supervised learning to aggregate signals across variable electrode layouts. The model outperforms baselines on neural decoding tasks, requires less training data, generalizes to new subjects, and reveals interpretable brain connectivity patterns.

**Strengths:**

- Validation across two types of neural recordings: EEG and intracranial EEG (iEEG)
- Thorough comparison against common aggregation methods and state-of-the-art models
- Strong ablations to test the importance of ensemble-wise loss, channel-wise loss, position encoding, and reconstruction loss

**Weaknesses:**

- In Tables 1, 2, and 3, consider adding statistical testing (e.g., Wilcoxon) between the best model and others. Then, correct p-values for multiple comparisons (e.g., Holm). Otherwise "We see significant improvements in performance" is not justified.
- I am confused about construction for " (1) ensemble-wise —the model determines if activities from two channel ensembles occurred consecutively, requiring an effective brain state representation at the ensemble-level". How do we know if activities from two channels occurred consecutively? In other words, could you explain how you construct pairs? How do you get the states?

**Questions:**

- Figure 1 has a short-time Fourier transform, but I did not find it to be discussed anywhere in the text.
- It would be great to see how much Gaussian fuzzing contributes to the final performance. EEG is spatially very smoothed.

---

> ### Author Response · Authors · 2024-11-22
>
> Thank you for taking the time to review our work\! We appreciate your feedback and questions, which we address below:
>
> 1. > In Tables 1, 2, and 3, consider adding statistical testing
>    - Done\! In tables 1 and 2, we use a Wilcoxon rank-sum test to compare between the first place and second place models in each section. For table 3, we use Dunnett’s test to pick out which ablations are significantly impactful.
> 2. > Could you explain how you construct pairs?
>    - Pairs consist of the activity from two different subsets of channels, each subset coming from a specific time point. The time points can either be consecutive times (positive pair) or randomly selected from any other time point (negative pair). Here, “consecutive” means “occurring 500ms afterwards”. In the case of positive pairs, we can ensure that the activities are consecutive by construction: we simply take activities from two windows, separated by 500ms. We clarify this in the text.
> 3. > short-time Fourier transform…did not find it to be discussed anywhere in the text.
>    - The STFT is part of BrainBERT’s preprocessing. We have revised the text to clarify this and explain how it fits into the broader pipeline.
> 4. > …how much \[does\] Gaussian fuzzing contribute to the final performance?
>    - Good question\! We’ve added an ablation experiment (see updated Tables 3 and 8). We find that Gaussian fuzzing provides small, but consistent benefits across decoding tasks.

---

> > ### Comment · Reviewer_jUF4 · 2024-11-25
> > **Rebuttal Response**
> >
> > I have increased the score to "6: marginally above the acceptance threshold". Thank you for addressing my concerns!

---

> > > ### Comment · Reviewer_jUF4 · 2024-11-27
> > > **Revised Response 2**
> > >
> > > After carefully reconsidering the strengths of the paper alongside the feedback from other reviewers, I have revised my decision and increased the score to “8: Accept, good paper.” The authors validate their model on diverse types of neural recordings (EEG and iEEG), compare its performance with multiple strong baselines, perform detailed ablation studies across various modules of the pipeline, and demonstrate the model’s interpretability capabilities. These strengths highlight the robustness and significance of the work.

---

> > > > ### Author Response · Authors · 2024-11-27
> > > >
> > > > We appreciate your consideration. Your suggestions and discussion were very useful in improving our paper's quality!

---

### Author Response · Authors · 2024-11-25

We sincerely thank all our reviewers for their time and dedication to reviewing our work. We are happy to hear that reviewers appreciate the work’s “strong rigor” (reviewer b3HQ), “strong empirical results” (reviewer VeNz), and anticipate that it will “benefit the neuroscience community at large” (reviewer vWCS). The majority of feedback requested that we sharpen our presentation, clean up phrasing, and add supplemental explanations. We have thoroughly reviewed each of our reviewer’s comments and addressed concerns (see below). The revised text now contains:
1. The statistical significance of our proposed model vs. baselines (Tables 1-3).
2. A clearer explanation of our interpretability results with respect to connectivity and attention (Figures 12-17, Algorithm 1)
3. Tests of our model’s performance over random subsets of electrodes (Figure 10)
4. Evaluations of the benefits of gaussian fuzzing with an ablation study (Table 3)
5. Evaluations of our pretraining scaling performance that are computed over all test subjects (Figure 7)
6. Evaluations of our model’s hold-one-subject out generalizability with additional temporal embeddings (TOTEM: Figure 11) and on an existing baseline (BrainBERT; Figure 6)

Additionally, we have updated the text (in teal) and codebase to provide more clarity for our work. We truly appreciate all the detailed feedback provided by our reviewers. We welcome any additional comments and feedback on our updated submission.

---

### Meta-Review · Area_Chair_u1MA · 2024-12-23

**Metareview:**

The paper introduces a self-supervised model called Population Transformer to model brain-wide neural activity sparsely and variably measured across subjects and datasets. Representations generated by this pre-trained model can then be used to perform downstream decoding tasks, leading to superior accuracy compared to models only trained on one specific dataset. The paper received overwhelmingly positive reviews, especially after the revision to improve the presentation and add additional evaluation and ablation components.

**Additional Comments On Reviewer Discussion:**

The paper benefited from the reviews and I comment the authors for addressing the weaknesses pointed out by the reviewers. There wasn’t a back and forth between reviewers and authors but this also wasn’t needed, given the strength of the paper.

---

### Decision · Program_Chairs · 2025-01-22

Accept (Oral)